# scAPAmod: Profiling Alternative Polyadenylation Modalities in Single Cells from Single-Cell RNA-Seq Data

**DOI:** 10.3390/ijms23158123

**Published:** 2022-07-23

**Authors:** Lingwu Qian, Hongjuan Fu, Yunwen Mou, Weixu Lin, Lishan Ye, Guoli Ji

**Affiliations:** 1Department of Automation, Xiamen University, Xiamen 361005, China; lingwuqian@foxmail.com (L.Q.); bridgehj@163.com (H.F.); monyunweng@163.com (Y.M.); 23220191151243@stu.xmu.edu.cn (W.L.); 2Xiamen Health and Medical Big Data Center, Xiamen 361008, China; yls@xmzsh.com

**Keywords:** single-cell RNA-seq, alternative polyadenylation (APA), Gaussian mixture model, patterns of APA usages

## Abstract

Alternative polyadenylation (APA) is a key layer of gene expression regulation, and APA choice is finely modulated in cells. Advances in single-cell RNA-seq (scRNA-seq) have provided unprecedented opportunities to study APA in cell populations. However, existing studies that investigated APA in single cells were either confined to a few cells or focused on profiling APA dynamics between cell types or identifying APA sites. The diversity and pattern of APA usages on a genomic scale in single cells remains unappreciated. Here, we proposed an analysis framework based on a Gaussian mixture model, scAPAmod, to identify patterns of APA usage from homogeneous or heterogeneous cell populations at the single-cell level. We systematically evaluated the performance of scAPAmod using simulated data and scRNA-seq data. The results show that scAPAmod can accurately identify different patterns of APA usages at the single-cell level. We analyzed the dynamic changes in the pattern of APA usage using scAPAmod in different cell differentiation and developmental stages during *mouse* spermatogenesis and found that even the same gene has different patterns of APA usages in different differentiation stages. The preference of patterns of usages of APA sites in different genomic regions was also analyzed. We found that patterns of APA usages of the same gene in 3′ UTRs (3′ untranslated region) and non-3′ UTRs are different. Moreover, we analyzed cell-type-specific APA usage patterns and changes in patterns of APA usages across cell types. Different from the conventional analysis of single-cell heterogeneity based on gene expression profiling, this study profiled the heterogeneous pattern of APA isoforms, which contributes to revealing the heterogeneity of single-cell gene expression with higher resolution.

## 1. Introduction

The cleavage and polyadenylation of pre-mRNA is an essential 3′ end processing mechanism for most eukaryotic mRNAs. Over the past two decades, genomic studies have revealed that most eukaryotic genes have multiple poly(A) sites, which can generate transcript isoforms encoding distinct proteins and/or with variable 3′ ends, through a mechanism called alternative polyadenylation (APA). Transcriptome profiling through direct 3′ end sequencing or RNA-seq has found up to 70% of genes in animals or plants undergo APA, which contributes significantly to transcriptome diversity and complexity. APA is extensively regulated in development and in a tissue-specific manner, and global changes in APA patterns have been observed in human diseases, including tumorigenesis and neuromuscular disorders, and during embryonic development and differentiation (reviewed in [1,2,3]). APA is a key layer of gene expression regulation, and APA choice is finely modulated in cells [4].

Advances in single-cell RNA-seq (scRNA-seq) have provided unprecedented opportunities to study APA in cell populations. Velten et al. designed a sequencing protocol with a prominent 3′ end sequencing bias and used a statistical pattern to characterize 3′ isoform choice variability in 48 single cells [5]. However, Velten’s approach is of very low sensitivity (~5%) and is severely affected by multiple rounds of amplification and batch effects [6]. Hwang et al. profiled four cell types in the *mouse* brain, which revealed that APA promotes protein diversity between different cell types and cellular states [7]. However, these sequencing projects are technically demanding, costly, and of low capture efficiency, hindering the widespread use of these technologies. Alternatively, computational approaches are emerging for profiling APA from scRNA-seq, particularly from those UMI-based protocols using 3′ enrichment in library construction, such as Drop-seq [8], CEL-seq [9], and 10× Genomics [10]. Some studies [11,12,13] characterized differential patterns of APA across tumor types from scRNA-seq data while analyzing data in a pseudo-bulk manner by pooling cells from the same cell type. Alternatively, a few computational tools, including APA-Seq [14], scAPA [15], Sierra [16], and scAPAtrap [17], were developed for identifying and quantifying APA sites in single cells from UMI-based scRNA-seq data. These pioneering studies focused on profiling differential APA usages between cell types or identifying APA sites. The diversity and pattern of cell-to-cell APA usages on a genomic scale among individual cells, especially in seemingly homogeneous cell populations, remains unappreciated.

scRNA-seq has demonstrated great potential to profile the cell-to-cell variability of genes across a population of single cells. So far, single-cell transcriptomic studies have been primarily limited to gene-level expression analysis, where the expression of each individual gene is the aggregation of isoforms derived from the same gene, to reveal cell-to-cell heterogeneity with distinct gene expression profiles and functional states [18] to investigate transcriptome stochasticity in response to internal or external signals [18] and to dissect tumor heterogeneity, to mention but a few. A few single-cell studies considered alternative splicing (AS) as an additional layer of information to interrogate the cell-to-cell variability driven by distinct AS events [18]. Using MISO, a tool originally developed for bulk RNA-seq, Shalek et al. identified AS events from scRNA-seq and revealed a bimodal, switch-like pattern in the expression of genes and splicing isoforms across 18 immune cells [19,20]. It was not until recently that computational approaches specifically designed for scRNA-seq, such as Expedition [21], BRIE [22] and SingleSplice [23] were developed, which continue to be used in investigating the variability of splicing variations at the single-cell resolution. Heterogeneity in splice isoforms between single cells from the same tissue or cell type has been found to be prevalent in single cells [5,19,20,21]. In contrast to the comparatively more fruitful studies on single-cell AS analysis, cases where single-cell transcriptome profiling is used to characterize the diversity of APA at the single-cell level are scarce. Very recently, Yang et al. characterized the cell-to-cell modality of APA for each neuron type in GABAergic neurons and found that the distal modality is predominant in different GABAergic neurons and the bimodality of APA of several genes could be utilized to distinguish GABAergic neuron subtypes [24]. However, there is no easy-to-use computational tool for exploring the cell-to-cell expression pattern of APA isoforms. Moreover, the preference of a particular poly(A) site (e.g., major/minor site or 3′ UTR/intronic site) from genes with multiple poly(A) sites in heterogeneous or homogeneous cell populations is also unexplored. Therefore, it is imperative to develop new computational methods for profiling APA modalities on a global scale from scRNA-seq to provide single-cell-level insight into APA heterogeneity and patterns of APA usages.

Here, we proposed an analysis framework based on a Gaussian mixture model, scAPAmod, to identify patterns of APA usage (also called modalities of APA) from homogeneous or heterogeneous cell populations at the single-cell level. We systematically evaluated the performance of scAPAmod using simulated data and scRNA-seq data. Moreover, we analyzed the dynamic changes in the pattern of APA usage using scAPAmod in different cell differentiation and developmental stages during *mouse* spermatogenesis and found that even the same gene has different patterns of APA usages in different differentiation stages. In addition, we analyzed cell-type-specific APA usage patterns and changes in patterns of APA usages across cell types. Different from the conventional analysis of single-cell heterogeneity based on gene expression profiling, this study profiled the heterogeneous pattern of APA isoforms, which contributes to revealing the heterogeneity of single-cell gene expression with higher resolution.

## 2. Results

### 2.1. Overview of scAPAmod

We developed an R package called scAPAmod for the identification of patterns (or modalities) of APA usages (Figure 1). To characterize the cell-to-cell variability of the APA usage of a gene or an APA site at the single-cell level, we defined three APA usage modalities: (i) unimodal means that the APA usage tends to be the same across all cells; (ii) bimodal means that there are two subpopulations with each one predominantly preferring a respective APA site; (iii) multimodal means that there are more than two subpopulations with different extents of APA usages.

First, we identified and quantified poly(A) sites from raw 3′ tag-based scRNA-seq data using existing tools such as scAPAtrap [17] and Sierra [16], which generates a poly(A) site expression matrix (Figure 1). Then, the matrix was transformed to an APA signal matrix, where each gene or each APA site was represented by the poly(A) usage index (PUI) [15] or ratio (Section 4). To generate a PUI matrix of APA genes, genes with at least two poly(A) sites were extracted and the PUI value of each APA gene was calculated, with each row representing an APA gene and each column representing a cell. Additionally, the PUI or ratio of an APA site could be calculated in the same way. Based on the PUI or ratio values for a gene or an APA site in a cell population, a variable number of components could be inferred by the Gaussian mixed model (GMM). Finally, the APA modality was distinguished according to the inferred component(s). Among the three APA modalities, unimodal means that PUI or ratio values of a gene or an APA site across all cells are from the same component, bimodal means that the PUI or ratio values consist of two components, and multimodal means that there are more than two components. Particularly, we identified APA modalities for 3′ UTR APA sites and non-3′ UTR APA sites. We also identified APA modalities for major and minor APA sites which were selected through their expression levels.

### 2.2. Evaluation of scAPAmod for APA Modality Identification

First, we used noise-free simulated data to evaluate the performance of scAPAmod (see Materials and Methods). Based on the number of poly(A) sites in the three cell types in the single-cell *mouse* spermatogenesis data, the number of cells in the simulated data was set to 100. According to the PUI value distribution of poly(A) sites in the spermatogenesis data, three groups were set for unimodal, five groups were set for bimodal, and seven groups were set for multimodal, resulting in a total of fifteen groups for the three modalities. Next, each group of data (i.e., a gene) was randomly generated 100 times, from which a 1500 × 100 noise-free simulation matrix was obtained. Then, we used scAPAmod for modality recognition. All the modalities in the simulated data were accurately identified using scAPAmod (Figure 2A). We also used ISOP and SAPAS to perform modality recognition on the same noise-free simulated data. Most patterns identified using ISOP were II patterns (37.6%, or bimodal) or V patterns (32.33%), while I patterns (13.4%, or unimodal), VI patterns (9.8%), or XI patterns (6.87%) were in the minority (Figure 2A and Appendix A). In contrast, most patterns identified using SAPAS were multimodal (53.33.6%) or bimodal (40%), with far fewer patterns being unimodal (6.67%) (Figure 2A and Appendix A). These results showed that ISOP and SAPAS tended to recognize more bimodal or multimodal patterns from the simulated data.

We also simulated data containing different degrees of missing values. With the increase in the proportion of missing values, the number of falsely identified modalities via ISOP greatly increased (Figure 2B). When 5% missing values were added into the simulated noise-free data, all modalities were predicted correctly using scAPAmod. When 10% to 20% missing values were added, only one item was falsely predicted using scAPAmod. When the missing rate was increased up to 50% to 60%, only eight items were falsely predicted. In contrast, ISOP wrongly predicted many multimodal items as bimodal no matter the degree of the missing rate. The degree of missing values had much less impact on SAPAS compared to ISOP; however, SAPAS falsely identified a considerable number of unimodal items as bimodal and multimodal with the increase in the degrees of missing values. This result demonstrated that scAPAmod could mitigate the impact of a high dropout rate and could effectively identify true modalities even in sparse data.

Next, we examined the consistency between the identified modalities via scAPAmod and ISOP and the true labels of the simulated data using the indicator of ARI (Adjusted Rand index) [25]. We calculated the ARI score using the true labels and the labels obtained using the two methods on different degrees of noise (0%, 5%, 10%, 15%, 20%, 25%, and 30%) (Figure 2C). For bimodal items, the ARI scores of scAPAmod and ISOP were both close to 1, which means that both methods could accurately identify the bimodal modality from the data. In contrast, for multimodal items, as the degree of noise increased, the ARI scores of both scAPAmod and ISOP decreased. However, the ARI scores of scAPAmod were consistently higher than those of ISOP, revealing that the labels predicted using scAPAmod was much closer to the true labels than those using ISOP. Next, to verify whether the ratio difference of the number of cells in two components of bimodal data would affect the accuracy of scAPAmod, ISOP, and SAPAS, we simulated bimodal data containing five different ratios of the number of cells in two components. The results showed that both scAPAmod and ISOP could identify correctly the bimodal pattern with a varied cell ratio of the two components (Appendix A). However, when there was a large difference in the number of cells in the two components of the bimodal, the accuracy of SAPAS in identifying bimodal was greatly reduced (Appendix A).

To further evaluate the scalability of scAPAmod for a higher number of cells, we constructed simulation datasets with 8000 cells using the same data generation procedure. Comparing the results to the 100-cell simulation data, we found that while the recognition performance of SAPAS increased slightly, the recognition performance of both scAPAmod and ISOP decreased in the 8000-cell noise-free simulation data (Appendix A vs. Figure 2A). However, the performance of scAPAmod was still significantly better than ISOP and SAPAS (Appendix A). With the increase in the proportion of missing values, the number of falsely identified modalities using either scAPAmod or ISOP increased (Appendix A). When 5% to 60% missing values were added into the simulated noise-free data, less than 100 multimodal items were falsely predicted as bimodal using scAPAmod. In contrast, ISOP wrongly predicted more unimodal and multimodal items as bimodal. Although SAPAS was little affected by the proportion of different missing values, when the degree of missing values increased, SAPAS still falsely predicted 200 multimodal items as bimodal. This result demonstrated again that scAPAmod could mitigate the impact of high dropout rate and could effectively identify true modalities even for sparse data with high sample sizes.

We calculated the ARI value on different degrees of noise (0%, 5%, 10%, 15%, 20%, 25%, and 30%) (Appendix A). For bimodal items, the ARI scores of scAPAmod and ISOP were both close to 1, which meant that both methods could accurately identify the bimodal from the data. In contrast, for multimodal items, the ARI scores of both scAPAmod and ISOP were not high. For bimodal items, the ARI scores of scAPAmod and ISOP were very close to 1, which was essentially the same as the results of the 100-cell simulation data. For the multimodal items, the ARI scores of scAPAmod and ISOP were much smaller than the results from 100-cell simulation data. Finally, we simulated bimodal data containing five different ratios of the number of cells in two components. The results showed that both scAPAmod and ISOP could identify correctly bimodal patterns with varied cell ratios in the two components. Similar to the results of the 100-cell simulation data, when there was a large difference in the number of cells in the two components of the bimodal pattern, SAPAS would not be able to accurately identify most bimodal patterns (Appendix A). Taken together, these results demonstrated the high scalability and performance of scAPAmod on data with different sample sizes and degrees of noise.

### 2.3. Modalities of 3′ UTR APA Sites in Mouse Sperm Data

Studies based on bulk RNA-seq have shown APA dynamics during *mouse* spermatogenesis differentiation [26,27]. Here, we attempted to explore the modalities of APA site usage based on single-cell *mouse* spermatogenesis data [15,28]. We used scAPAtrap to identify poly(A) sites from the scRNA-seq data. Across the three cell types of the single-cell *mouse* spermatogenesis data, the overall distributions of the number of poly(A) sites expressed in all cells were similar (Appendix A). Moreover, most of the poly(A) sites were not expressed in the cells due to the inherent high sparsity of scRNA-seq.

We checked the heterogeneity of poly(A) site expressions through the coefficient of variation of each poly(A) site and each gene in all cells. A total of 99% of cells (2028/2042) were more heterogeneous at the APA level than at the gene level (Appendix A). For example, for the *Mdm2* gene [26], the original gene expression was similar across all cell types, while the usage rate of APA denoted by the PUI score varied across different cell types (Figure 3A).

Next, we used scAPAmod to analyze whether there were patterns of heterogeneity of poly(A) site usages. We used the PUI indicator to represent the usage of the proximal poly(A) site in an APA gene. In each stage of cell differentiation, most genes worked in the unimodal modality, which means that the APA usages of the same gene were similar among cells from the same stage (Figure 3B). Moreover, the distributions of modalities of APA usages in the process of cell differentiation were consistent, with the number from high to low being: unimodal, bimodal, and multimodal.

The *Mdm2* gene was identified as bimodal using scAPAmod. The two components (C1 and C2) of the *Mdm2* gene determined using scAPAmod could clearly show poly(A) site expression differences among different cells (Figure 3C(I)). Moreover, the PUI distribution of the gene showed that the average value of one component (C1) was less than 0, and the average value of the other component (C2) was greater than 0 (Figure 3C(II)). We also examined the expression distribution of the two poly(A) sites of the *Rpl34* gene and *Fau* gene (Appendix A). The *Rpl34* gene belonged to the multimodal modality, and the expression distribution of the two poly(A) sites of this gene could be divided into three components. The average PUI values of the three components of the gene were close to 0, between 0 and 1, and greater than 1, respectively (Appendix A). The gene expression of *Rpl34* was different in different cells. The *Fau* gene belonged to the unimodal modality, and all cells had high expression at one poly(A) site and low expression at the other poly(A) site. The average PUI value of the *Fau* gene was distributed between −1.5 and −1.0 (Appendix A).

We also used scAPAmod to identify and analyze the usage patterns of poly(A) sites with high and low usage rates across different cell types. The distribution of the usage patterns of the two types of poly(A) sites was not the same (Figure 3D). In the SC and RS cell types, the usage patterns of major poly(A) sites in the 3′ UTR were mostly bimodal, followed by multimodal, and the fewest were unimodal. In the ES cell types, the usage patterns of most major poly(A) sites in the 3′ UTR was bimodal, but the fewest were multimodal. The distribution of the usage patterns of major poly(A) sites in the non-3′ UTR was also different. In the SC cell type, most major poly(A) sites belonged to the multimodal modality, followed by the bimodal modality. In the RS and ES cell types, the number of major poly(A) sites belonging to the bimodal modality was the largest. In the RS cell types, the fewest were unimodal. In the ES cell types, the fewest were multimodal. For minor poly(A) sites, either in the 3′ UTR or the non-3′ UTR, most were unimodal in all the three cell types, followed by bimodal, and the fewest were multimodal.

Next, we examined the genomic regions with major poly(A) sites and the minor poly(A) sites. We found that major poly(A) sites were mainly located in the 3′ UTR, with a small number being in introns. However, minor poly(A) sites were mainly distributed in introns, with a small number located in 3′UTR and coding regions. We also examined the usage patterns of poly(A) sites of specific gene regions in non-3′ UTRs. The distribution of usage patterns of major poly(A) sites in different regions was different (Appendix A). However, minor poly(A) sites had the same distribution of usage patterns regardless of the region where the poly(A) sites were located. The two types of poly(A) sites of non-3′ UTRs were mainly located in introns, followed by the coding region. However, the usage patterns of two types of poly(A) sites were rarely detectable in the 5’ UTR and the exon region. These results indicated that the heterogeneity of poly(A) sites not only existed in the 3′ UTR, but also in the non-3′ UTR, especially in introns and coding regions.

We used the coefficient of variation (CV) to analyze the variation in usages of major and minor poly(A) sites in the three cell types based on their expression levels (Appendix A). In SC, the mean CV values of major poly(A) sites and minor ones were 0.13 and 1.29, respectively. The CV values in RS were similar to the SC (major = 0.12 and minor = 1.33). In ES, the mean CV of major sites was similar to SC or RS (0.13), while the mean CV of minor sites was smaller (0.85). This result indicated that the heterogeneity of expression of minor poly(A) sites was higher than that of major ones.

### 2.4. Cell-Type-Specific APA Modalities

In order to further analyze the heterogeneity of poly(A) sites, we examined the changes in the modalities of the same gene in different cell types. Among the three cell types, the number of the usage patterns of poly(A) sites were unimodal, bimodal, and multimodal in descending order, and the number of genes whose usage patterns was detected gradually decreased during the differentiation of cells (Figure 3B and Figure 4A). Fourteen genes were common among the three cell types; more genes (30) were exclusively identified with modalities in SC than in RS (10) or ES (13) (Figure 4A). More genes (22) were identified with modalities in both SC and RS than in RS.

In order to study the usage patterns changes in genes and their functional effects during cell differentiation, we divided the single-cell *mouse* spermatogenesis data into two cell differentiation stages: SC-RS and RS-ES. We counted the usage pattern changes in genes in the two cell differentiation stages separately. They were divided into four situations: same, change, appear, and disappear. Same means that the gene usage pattern did not change in the process of cell differentiation. Change means that the gene usage pattern changed in the process of cell differentiation. Appear means that the usage pattern of the gene changed from unrecognizable to recognizable during cell differentiation. Disappear means that the usage pattern of the gene changed from recognizable to unrecognizable during cell differentiation. The statistical results showed that 79 genes had detectable usage patterns in the differentiation stage of SC-RS, and 62 genes had detectable usage patterns in the differentiation stage of RS-ES (Figure 4B). In the cell differentiation stage of SC-RS, the usage patterns of 38% of genes disappeared, the usage patterns of 33% of genes remained unchanged, the usage patterns of 13% of genes changed, and the usage patterns of 16% of genes appeared. In the cell differentiation stage of RS-ES, the usage patterns of 52% of genes with detectable usage patterns disappeared, the usage patterns of 21% of genes with detectable usage patterns appeared, the usage patterns of 8% of genes changed, and the usage patterns of the remaining 19% of genes were maintained. This meant that with the differentiation of sperm cells, the number of genes whose usage patterns could be detected was gradually decreasing. The usage pattern changes in the same gene across different stages decreased.

Next, in order to check the specific usage pattern changes in different cell differentiation stages, we also further statistically analyzed the usage pattern of genes whose usage patterns changed in the two cell differentiation stages. It was found that during the SC-RS stage, the usage patterns of 10 genes changed, of which 5 genes changed from bimodal to unimodal, and 4 genes changed from unimodal to bimodal. Overall, 60% of genes changed from other usage patterns to unimodal. In the RS-ES stage, the usage patterns of five genes changed. Among them, the usage patterns of three genes changed from bimodal to unimodal, and two genes changed from unimodal to bimodal. Therefore, the distribution of usage patterns of poly(A) sites across the three cell types tended to be the same, with most being unimodal, followed by bimodal, and the fewest being multimodal (~0) (Appendix A). Particularly, there were 14 genes that were recognized usage patterns in the three cell types, and of these, 8 genes changed usage patterns. We took three genes as an example. The *Srpk1* [29] gene is highly specific to protein phosphorylation. The expression distribution of this gene was different in different cell types, and the usage pattern changed from unimodal to bimodal with cell differentiation (Figure 4C). The targeted disruption of the *Ppplcc* [30] gene can cause male infertility in mice due to impaired spermiogenesis. The usage pattern of this gene changed from bimodal to unimodal with cell differentiation (Appendix A). The *Rpl34* gene is related to diseases and can affect growth retardation, hereditary bone marrow failure syndrome, congenital abnormalities, malignant tumors, anemia, pancytopenia, etc. The usage patterns of this gene changed from multimodal to unimodal with cell differentiation (Appendix A).

### 2.5. Non-3′ UTR-APA Modality

We also used the data of poly(A) sites of the non-3′ UTR of single-cell *mouse* spermatogenesis to perform usage patterns recognition. We calculated the relationship between genes with usage patterns in the 3′ UTR and non-3′ UTR. The usage patterns of 69 genes were detected using scAPAmod from 3′ UTR-APA sites’ data. Additionally, the usage patterns of 351 genes were detected using scAPAmod from non-3′ UTR-APA sites’ data. When comparing the two results, it was found that 23 genes with usage patterns were both detected in the two regions, most of the genes were in SC cell types, and no genes were detected with usage patterns in ES cell types (Appendix A).

We counted the number of non-3′ UTR-APA sites with usage patterns in the three cell types, respectively. Additionally, we found that the patterns of non-3′ UTR-APA sites’ usages were similar across different cell types (Figure 5A). Among the three cell types, the usage patterns of most non-3′ UTR-APA sites were unimodal, followed by bimodal. This meant that whether it was in the 3′ UTR or the non-3′ UTR, the distributions of usage patterns of the poly(A) sites were consistent. It could be seen from the statistics results that there were 46 unique genes in SC cell types, 25 unique genes in RS cell types, and 99 unique genes in ES cell types. There were 252 genes shared by the SC and RS cell types alone, of which, 58 genes with usage patterns changed. The RS and ES cell types shared 305 genes alone, of which, 29 genes with usage patterns changed (Appendix A).

We also counted the pattern changes in APA sites’ usages in the differentiation stages of SC-RS and RS-ES (Figure 5B), which were also divided into four situations: same, change, appear, and disappear. It could be seen from the statistical results that in the SC-RS stage, the usage patterns of about 41% of the poly(A) sites’ usages did not change, and the usage patterns of 19% of the poly(A) sites changed. In addition, 21% of the poly(A) sites’ detected usage patterns with differentiated cells. As the cells differentiated, 19% of poly(A) sites’ usage patterns could no longer be detected. In the RS-ES stage, about 33% of poly(A) sites’ usage patterns were detected. The usage patterns of 30% of poly(A) sites did not change. In addition, 27% of poly(A) sites’ usage patterns could no longer be detected. The usage patterns of 10% of poly(A) sites changed.

In order to further study the pattern changes in poly(A) sites’ usages, we made statistics on the various situations that may occur at each stage of cell differentiation (Figure 5C). In the cell differentiation stage of SC-RS, the usage patterns of 62% of the poly(A) sites changed from bimodal to unimodal; the usage patterns of 31% poly(A) sites changed from another pattern to bimodal, of which, 29% of poly(A) sites changed from unimodal to bimodal; and only 2% of poly(A) sites changed from multimodal to bimodal. The usage patterns of 5% of poly(A) sites changed from multimodal to unimodal. In the cell differentiation stage of RS-ES, the usage patterns of 53% of poly(A) sites changed from bimodal to another pattern, of which, 50% of poly(A) sites changed from bimodal to unimodal, and only 3% of poly(A) sites changed from bimodal to multimodal. The usage patterns of 37% of poly(A) sites changed from unimodal to bimodal. Additionally, the usage pattern of only 3% of poly(A) sites changed from multimodal to unimodal. This result was consistent with the statistical results of the usage patterns of each cell type. Most usage patterns were unimodal in each cell type in the non-3′ UTR.

It could be seen from the statistical results that 93 genes with usage patterns were detected in all three cell types. Among them, the usage patterns of 45 genes changed (Figure 5E, Appendix A). We searched for the functions of these genes and found that 14 of them have special functions: *Actb* [31], *crem* [32], *Ddx5* [33], *Cetn1* [34], *Spa17*, *Hsp90aa1* [35], *Pabpc1* [36], *Ppp1cc*, *Srpk1*, *Tcp11* [37], *Ubb* [38], *Ybx1* [39], *Gkap1* [40], and *Gpx4* [41]. *Crem*, *Ddx5*, *Pabpc1*, *Cetn1*, *Ppp1cc*, and *Srpk1* are related to polyadenylation. We examined the density distribution of the PUI value of these six genes in turn. Here, taking the density distribution of the PUI data of the *Crem* gene as an example (Figure 5D), it showed that the data distribution of the two poly(A) sites of the *Crem* gene were similar in different cell types. It belonged to the unimodal modality in SC and ES and belonged to the bimodal modality in RS.

Similarly, we also examined the genes *Ddx5*, *Pabpc1*, *Cetn1*, *Ppp1cc*, and *Srpk1* (Appendix A). This showed that the usage pattern of the *Ddx5* gene and the *Ppp1cc* gene changed from bimodal to unimodal with cell differentiation. The usage pattern of the *Pabpc1* gene changed from unimodal to multimodal during cell differentiation. The usage pattern of the *Cetn1* gene changed from unimodal to bimodal and finally became unimodal with cell differentiation. The usage pattern of the *Srpk1* gene changed from unimodal to bimodal with cell differentiation.

## 3. Discussion

With the development of scRNA-seq technology, many works have been devoted to studying the heterogeneity of gene expression at the single-cell level. However, these works are mainly used for gene-level expression analysis or for cell types to be explored in a small number of cells. There are few studies on poly(A) sites’ recognition and dynamic analysis based on the scRNA-seq technology. APA is one of the key regulatory factors of gene expression, which is involved in cell development, proliferation, differentiation, and other processes as well as the occurrence of diseases. Therefore, it is meaningful to analyze the diversity and pattern of poly(A) expression at the single-cell level.

We proposed a new computing framework, scAPAmod. Based on the Gaussian mixture model, the framework is used to recognize the usage patterns of single-cell poly(A) sites. Using the single-cell *mouse* spermatogenesis data [15,28], we used the scAPAmod framework to analyze the dynamic changes in patterns of APA usages across different genomic regions and cell differentiation and development stages. It was found that even the same gene had different usage patterns in the different cell differentiation stages (Figure 3C and Appendix A). In addition, we also analyzed the preference of APA usage patterns in different genomic regions. It was found that the usage patterns of poly(A) sites of the same gene in 3′ UTR and non-3′ UTR were different (Figure 3D). This study analyzed the heterogeneity of APA usage patterns at the level of APA transcripts, and it could reveal the heterogeneity of single-cell gene expression with a higher resolution. Further, in order to examine whether genes with APA modalities can delineate known cell type clusters, we used Seurat V4.0.2 (dim.use = 4) to cluster cell types based on the APA expression profile of genes with APA modalities. A total of 420 genes with 3′UTR-APA or non-3′UTR-APA modalities were used. The mean silhouette width and ARI score of the clustering result based on the APA profile was 0.71 and 0.92, respectively (Appendix A). In contrast, the mean silhouette width and ARI score of the clustering result based on the gene expression profile of all genes was 0.80 and 0.57, respectively (Appendix A). The result based on genes with APA patterns was comparable to that based on whole-gene expression profiles, but the feature space for clustering was greatly reduced (420 vs. 22,032 genes). This result preliminarily suggested that the APA expression profile of genes with APA modalities was sufficient to distinguish cell types and indicated the potential use of our scAPAmod tool for identifying important features for cell type clustering.

We compared the performance of scAPAmod with ISOP and SAPAS using simulated data. The simulation results showed that scAPAmod could accurately identify the usage patterns of almost all poly(A) sites, while ISOP or SAPAS recognized the usage patterns of most poly(A) sites as bimodal or multimodal (Figure 2A and Appendix A). The simulation results for seven different ratios of NA values showed that as the ratio of NA values increased, the pattern recognition error rate of ISOP greatly increased. However, the number of recognition errors of scAPAmod or SAPAS was much smaller than that for ISOP (Figure 2B). The simulation tests under different noise levels (5~30%) showed that as the noise data increased, the three tools could still accurately identify bimodal modalities. The recognition performance of scAPAmod for multimodal modalities did not perform well when adding 15% noise. However, the recognition performance of ISOP for multimodal modalities was always poor (Figure 2C). The performance advantages of scAPAmod are mainly due to the following reasons: First, we used a new components correction method based on the BIC score, which could achieve the correct division of the components of the Gaussian mixture model. Second, we used the combination of the EM algorithm and K-Means algorithm to optimize the parameter values *μ_i_* and Σ*_i_*, which could avoid the results from falling into the local optimum effectively, thereby ensuring the accuracy of component division. Of note, in this study, we did not impute missing data but removed cells with missing values before identifying modalities. This is because that there are no dropout imputation methods available for the imputation of APA signals. Although a recent tool, scDaPars [42], was proposed for identifying and imputing APA signals from scRNA-seq, the scDaPars pipeline identified APA dynamics using DaPars, the principle of which was different from the tools used in our study, scAPAtrap or Sierra. Therefore, we did not consider imputation in this study. However, scAPAmod is flexible to incorporate the imputation step into its framework when dedicated imputation tools are available for recovering APA signals from scRNA-seq.

We used scAPAmod to identify and analyze the patterns of 3′ UTR poly(A) sites’ usages on the single-cell *mouse* spermatogenesis data [15,28]. The results showed that the number of detected genes with usage patterns decreased overall as cells differentiated. In each stage of cell differentiation, the overall distribution of usage patterns of genes was consistent, with the number from high to low being unimodal, bimodal, and multimodal (Figure 3B). This meant that the distribution of the usage rate of most genes was relatively concentrated among different cells. The usage rates of only a few genes were widely distributed, showing heterogeneity among different cells. Through the identification and analysis of the usage patterns of genes in the 3′ UTR, it was found that there were 14 genes with recognized usage patterns shared across the three cell types, and the usage patterns of 8 genes were changed.

We also analyzed the usage patterns of major poly(A) sites and minor poly(A) sites in the 3′UTR and non-3′ UTR across different cell types. It could be found from the results that the usage patterns of major poly(A) sites in different regions were different. However, regardless of whether the minor poly(A) sites were in the 3′ UTR or non-3′ UTR, the usage patterns’ distribution was consistent, with the largest number of unimodal, the middle number of bimodal, and the smallest number of multimodal patterns (Figure 3D). The data distribution of minor poly(A) sites was more volatile than major APA (Appendix A). Through the identification and analysis of the usage patterns of major poly(A) sites in the non-3′ UTR, 93 genes with usage patterns were detected across three cell types. Among them, the usage patterns of 45 genes were changed (Figure 5E, Appendix A). Among them, six genes were related to polyadenylation. Two genes were detected in both 3’UTRs and non-3’UTRs: *Ppp1cc* and *Srpk1*. The *Srpk1* [29] gene is highly specific to protein phosphorylation. The targeted disruption of the *Ppplcc* [30] gene can cause male infertility in mice due to impaired spermiogenesis. The usage patterns of the two genes were changed in different cell types. The results showed that scAPAmod could reveal the heterogeneity of gene expression with the single-cell resolution.

This study focused on the APA modality rather than gene expression or APA dynamics analyzed in many other single-cell studies [11,15,16,17]. Although methods such as scDaPars [42] also investigated APA in single cells, they focused on profiling APA dynamics between cell types or identifying APA sites, which is different from the identification of APA modalities in this study. APA modality represents the pattern of the cell–cell heterogeneity of APA usages in a cell population, which is different from APA dynamics which considers the differential use of APA sites between two cell populations (e.g., cell types) or two cells. The number of genes identified with an APA modality was relatively low in this study (Figure 4), which may be due to the following reasons: First, in many other single-cell studies on APA [11,15,16,17], cells from the same cell type were pooled for the analysis of the APA site switching between cell types. Differently, the APA modality was identified from heterogenous single cells, and the expression (or APA) profile of an individual cell was much sparser than the pooled cell populations. Second, we used the GMM model to identify APA modalities, which required that the APA usages in subpopulations presented distinct components. Here, we also tried to use scAPAmod to analyze *Human* PBMC 4k data [43]. The gene–cell expression matrix of this dataset contains 33,694 genes in 4340 cells. After the identification of poly(A) sites from the raw scRNA-seq using scAPAtrap, 15,304 poly(A) sites were identified in 3′ UTRs, and 3170 3′ UTR-APA genes were obtained. We considered three main cell types for identifying APA modalities: T cells (2308 cells), B cells (612 cells), and Monocytes (1164 cells). Only a small number of APA genes with modalities were detected—13 genes in T cells, 11 genes in B cells, and 17 genes in Monocyte cells. Among these genes, unimodal and bimodal patterns were detected, while no multimodal patterns were detected. The relatively low number of APA modalities present in the PBMC data may be due to the following reasons: First, the APA profile obtained from the PBMC data was sparse, even sparser than the *mouse* spermatogenesis data we analyzed; consequently, there were not sufficient cells for modality identification. Second, the APA dynamics among cell types in PBMCs are not as distinguishable as those during spermatogenesis, and the impact of APA regulation in PBMCs is not as significant as that during the spermatogenesis. Third, we used the GMM model to identify APA modalities, which required that the APA usages in subpopulations presented distinct components. Therefore, our results were conserved, and the number of genes with APA modalities may have been underestimated. However, users can inspect each single gene with an APA modality identified by our method and may find genes with biological importance. One future research direction is to improve the prediction sensitivity of our tools on small sample data or highly sparse data. At the same time, with the progress of sequencing technology and the increasing amount of data, our method will help researchers identify APA modalities in more APA sites and/or genes.

## 4. Materials and Methods

### 4.1. Data

The single-cell *mouse* spermatogenesis data from 10× Chromium [15,28] was used in this study, which contain three cell types according to their differentiation stages: the first stage is the differentiation of cells into spermatocytes (SC, 693 cells); the second stage is the differentiation of spermatocytes into round sperm cells (RS, 1140 cells); the third stage is differentiation into mature sperm cells (ES, 209 cells). We used scAPAtrap [17] to identify and quantify poly(A) sites in each single cell, following the same pipeline as in the study of scAPAtrap for data preprocessing and poly(A) site identification. We obtained 43,395 poly(A) sites in 3′ UTRs (including the 3′ UTR extension region within 1000 bp downstream) and 70,108 sites in non-3′ UTRs. We also obtained the expression level represented by the count of unique molecular identifiers (UMIs) of each poly(A) site.

### 4.2. Calculation of APA Index

We used the index of PUI to quantify the APA usage of an APA gene in single cells, which has been adopted in the previous study for single-cell APA analysis [15]. Given an APA site *i* in an APA gene, the PUI in a cell is:(1)$${\mathrm{PUI}}_{i}=log_{2}\left(\frac{pc_{i}+1}{e^{mean\left(log\left(pc+1\right)\right)}}\right)$$
where *pc_i_* represents the expression of the *i*-th poly(A) site, and *pc* represents the sum of the expression of all poly(A) sites in the same gene. If no poly(A) site is expressed in this gene, the PUI value is NA.

For APA genes containing more than two 3′ UTR poly(A) sites, the one closest to the stop codon is the proximal site and the one closest to the 3′ UTR end is the distal one. We used the PUI of the proximal poly(A) site to characterize the overall usage of this gene. PUI < 0 means below-average usage of the proximal poly(A) site of the gene, i.e., longer 3′ UTR; PUI > 0 means above-average usage of the proximal site, i.e., shorter 3′ UTR; PUI = 0 means the average usage of proximal poly(A) sites among all sites in the gene.

In addition to PUI, we also used the ratio to represent the usage of an APA site:(2)$$ratio_{i}=\frac{pc_{i}}{pc}$$

### 4.3. Identification of Modalities of APA Usage

We adopted the Gaussian mixture model to describe the distribution of APA usages of a gene in a group of cells. Assuming that there is a latent component (cell population) denoted by a latent variable *z*, the corresponding marginal probability distribution can be obtained:(3)$$P\left(\eta |\theta \right)=\overset{M}{\underset{i}{\sum }}P\left(\eta ,z=C_{i}|\theta \right)=\overset{M}{\underset{i}{\sum }}p_{i}\phi \left(\eta |\mu_{i},\Sigma_{i}\right)$$
where *η* represents the PUI value of a gene or a poly(A) site, or the ratio of a poly(A) site. *M* represents the number of components. *C_i_* represents the *i*-th component. The probability value of each cell belonging to the *C_i_* component is *p_i_*; *θ* represents the parameter of the marginal probability distribution; *μ_i_* indicates the mean PUI of the component *C_i_*; Σ*_i_* is the covariance of PUI values in component *C_i_*. If there are *i* components, *i* Gaussian distributions ϕ will be constructed correspondingly.

In this study, we set *M* to 3, which means that one to three models were constructed to calculate the marginal probability distributions. Moreover, we calculated the BIC (Bayesian information criterion) value for each model according to the BIC criterion. The smaller the BIC value, the more accurate the component division was. The pseudocode of scAPAmod is shown in Algorithm 1. In order to avoid affecting the accuracy of component division due to the close BIC values of the three patterns, we further corrected the optimal number of components. First, we removed the largest BIC value among the three BIC values. Then, we compared the remaining two BIC values and calculated the ratio of the smaller BIC value to the larger BIC value. If the ratio greater than 0.95 meant that the two BIC values were very close, we would reduce the number of components. Next, if the number of components was greater than 1, the component with less than 10 cells would be removed. We combined the EM (Expectation Maximum) algorithm and K-Means algorithm to continuously optimize the parameter values of *μ_i_* and Σ*_i_*. This method could effectively avoid the local optimum of the result, thereby ensuring the accuracy of component division.
**Algorithm 1** The pseudocode of scAPAmod**Input:** PUI value of a gene or a poly(A) site, or the ratio of a poly(A) site *η*Step1. BIC(1), BIC(2), BIC(3) ← Optimal_Clusters_GMM (η, 3, criterion=“BIC”);Step2. remove the largest BIC value among the three BIC values;Step3. compare the remaining two BIC values, recorded smaller BIC value as BIC(s) and larger BIC value as BIC(l);Step4. if BIC(s)/BIC(l)>0.95       then M=min(s,l)else M=sStep5. for i ← 1 to M      do if the number of cells of i-th component < 10then M=−1**Output:** number of components *M*Note: BIC(K) represents the BIC value corresponding to the Gaussian mixture model obeying K Gaussian distributions

We used the Optimal_Clusters_GMM function in the ClusterR [44] package to calculate the BIC value. The GMM function is used to build the Gaussian mixture model. Additionally, the predict_GMM function was used to output the result of the model.

### 4.4. Identification of 3′ UTR-APA Modality

In order to identify the usage patterns of 3′ UTR-APA sites, we selected poly(A) sites with high expression for analysis. Specifically, we filtered out poly(A) sites expressed in less than a quarter of cells or cells with less than one-tenth of expressed poly(A) sites. Only genes with at least two 3′ UTR poly(A) sites were used for subsequent APA modality analysis. For 3′ UTR-APA genes, we used the PUI of the proximal poly(A) site (Equation (1)) as the proximal APA usage of the gene to represent its relative 3′UTR length. Then, the APA modality of each APA gene was identified using PUI values in single cells. In addition, we also analyzed the APA modality of major and minor 3′ UTR sites, which were determined based on the expression level (i.e., read counts) of APA sites in a gene. For 3′ UTR-APA sites, the poly(A) site with the largest expression was considered as the major one (major APA), and the poly(A) site with the smallest expression was considered as the minor one (minor APA). However, for major and minor APA sites, we used ratio rather than PUI to quantify the usage of the major or minor site. This is because the major and minor site can be any poly(A) site in an APA gene. If we calculated the PUI value, the PUI of the major site would always be high and the minor low. For genes with two poly(A) sites, the APA modality of these genes based on PUI values would always be the same (unimodal). Therefore, we used ratio instead to calculate the usage of the major/minor site (Equation (2)). Here, the ratio value was log2 transformed with a pseudo number 1, which kept the value range of ratio and the value range of PUI in the same order of magnitude. Then, the usage pattern of major or minor site was obtained using scAPAmod using the log2 transformed ratio values as input.

### 4.5. Identification of Non-3′ UTR-APA Modality

For the identification of non-3′ UTR-APA usage patterns, we first selected the genes containing more than three poly(A) sites (at least one non-3′ UTR poly(A) site and at least two 3′ UTR sites). If there were multiple poly(A) sites of a gene in non-3′ UTRs, only the poly(A) site with the largest expression was selected. Next, we selected poly(A) sites expressed in at least half of the total cells and cells with more than half expressed poly(A) sites. We calculated the PUI of each APA site and used scAPAmod to identify the APA modality.

### 4.6. Performance Evaluation

There are many tools, such as APA-Seq [14], scAPA [15], Sierra [16], and scAPAtrap [17], for identifying poly(A) sites from scRNA-seq and/or detecting differential APA usages among cell types, but these tools are not capable of profiling single-cell APA modalities in homogeneous or heterogeneous cell populations. Previously, SAPAS [24] utilized 3′-tag-based scRNA-seq data to identify novel poly(A) sites and quantify APA at the single-cell level. In SAPAS, the desired distributions of different modalities were explicitly defined, e.g., distal = c(0, 0, 1), proximal = c(1, 0, 0), middle = c(0.1, 0.8, 0.1), etc., and the modality of the closest reference distribution was selected as the modality of the gene’s poly(A) site usage. In addition, ISOP [45] was developed for characterizing expression patterns of isoform pairs of the same gene at the single-cell level from single-cell isoform-level expression data. Although ISOP was not purposely designed for APA analysis, it can in principle be used to identify different patterns of APA in a group of cells, using the PUI values of a gene in individual cells as the input. We compared scAPAmod with SAPAS and ISOP in this study.

We used simulated data to evaluate the performance of scAPAmod. For unimodal patterns, we constructed three groups of randomly generated data following one Gaussian distribution with mean values of 0, >0, and <0, respectively. For bimodal patterns, we constructed five sets of mixed data following two Gaussian distributions with the mean values of the two components being >0 and <0, >0 and >0, < 0 and <0, >0 and ~0, and <0 and ~0. For multimodal patterns, we constructed seven sets of mixed data following three Gaussian distributions with the mean values of the three components being >0 and <0 and ~0, >0 and >0 and >0, <0 and <0 and <0, >0 and >0 and ~0, <0 and <0 and ~0, >0 and >0 and <0, and <0 and <0 and >0. Here, the variance of the Gaussian distribution was set as 0.1, and we set the mean value according to the single-cell *mouse* spermatogenesis data. For unimodal patterns, the mean values for the three groups were 2 (>0), −2 (<0) and 0 (~0). For bimodal patterns, the mean values for the two components in the five groups were: (2, −2), (2, 0.5), (−2, −0.5), (2, 0), and (−2, 0). For multimodal patterns, the mean values for the three components in the seven groups were: (2, −2, 0), (2, 1, 0.5), (−2, −1, −0.5), (2, 1, 0), (−2, −1, 0), (2, 1, −2), and (−2, −1, 2)

For the 15 groups of data with the above three patterns, we iterated them 100 times to obtain noise-free data (1500 poly(A) sites × 100 cells). Then, we used scAPAmod, ISOP [45], and SAPAS [24] to identify patterns for each of the 15 groups. We considered I pattern of ISOP as unimodal, II and V patterns as bimodal, and VI, X, and XI patterns as multimodal. We considered the distal, proximal, and middle pattern of SAPAS as unimodal, the bimodal pattern as bimodal, and the multimodal pattern as multimodal.

In order to further test the consistency between actual labels and labels predicted using scAPAmod, we adopted an evaluation index, ARI. Moreover, we added different degrees of noise (5%, 10%, 15%, 20%, 25%, and 30%) to the noise-free simulated data and calculated ARI values. Since SAPAS did not have the function of cell clustering, here, we only compared scAPAmod and ISOP.

We also simulated the data containing various degrees of missing values. It was necessary to ensure that each component in the data of the three patterns contained more than 10 cells, so the proportion of missing values was set to 5%, 10%, 20%, 30%, 40%, 50%, and 60%, respectively.

In order to further investigate the accuracy of scAPAmod for bimodal patterns with different numbers of cells for two components, we constructed five groups of data with different ratios of the two components belonging to bimodal. Among them, the ratios of the two components were set to 5:5, 6:4, 7:3, 8:2, and 9:1. The means of them were set to 10 and −10, and the variances were both set to 0.1.

## Figures and Tables

**Figure 1 ijms-23-08123-f001:**
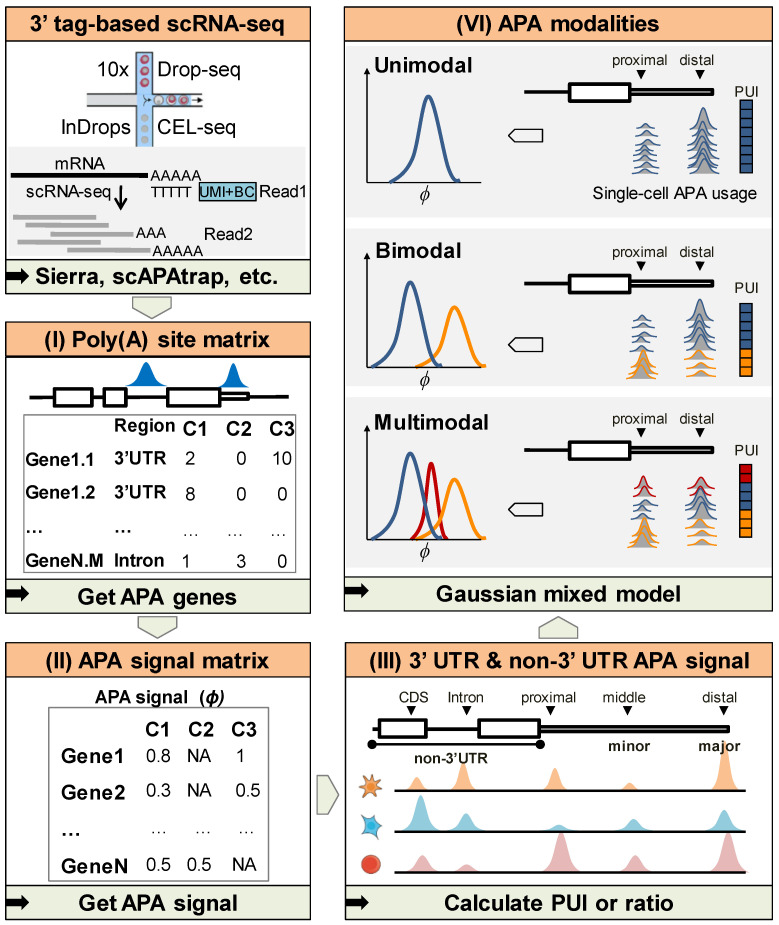
The flow chart of APA modality recognition using scAPAmod. (I) Poly(A) sites were identified and quantified from raw 3′tag-based scRNA-seq data using existing tools. (II) The poly(A) site matrix was transformed to an APA signal matrix, where each gene or each APA site was represented by the poly(A) usage index (PUI) or ratio. (III) PUI or ratio of 3′ UTR APA sites and non-3′ UTR APA sites could be calculated for each gene or poly(A) site. (VI) Given the PUI or ratio values for a gene or an APA site in a cell population, the APA modality could be inferred by the Gaussian mixed model.

**Figure 2 ijms-23-08123-f002:**
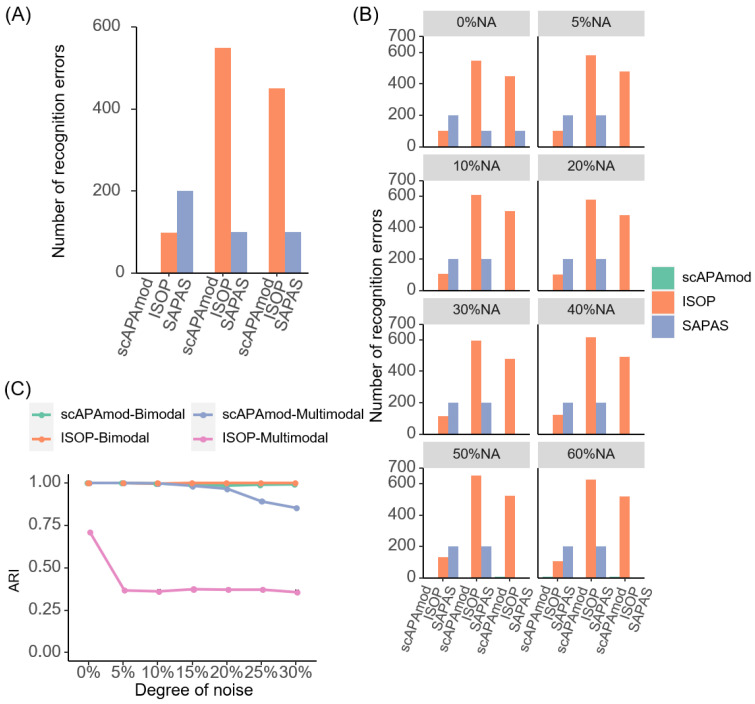
Simulation results of the 100-cell simulation data. (**A**) Number of recognition errors for different modalities using noise-free simulated data. (**B**) Number of recognition errors for different modalities using the simulated data with different degrees of missing values. (**C**) ARI of simulated data with different degrees of noise. Here, SAPAS was not included as it only outputs modality without information of cells in the identified modality.

**Figure 3 ijms-23-08123-f003:**
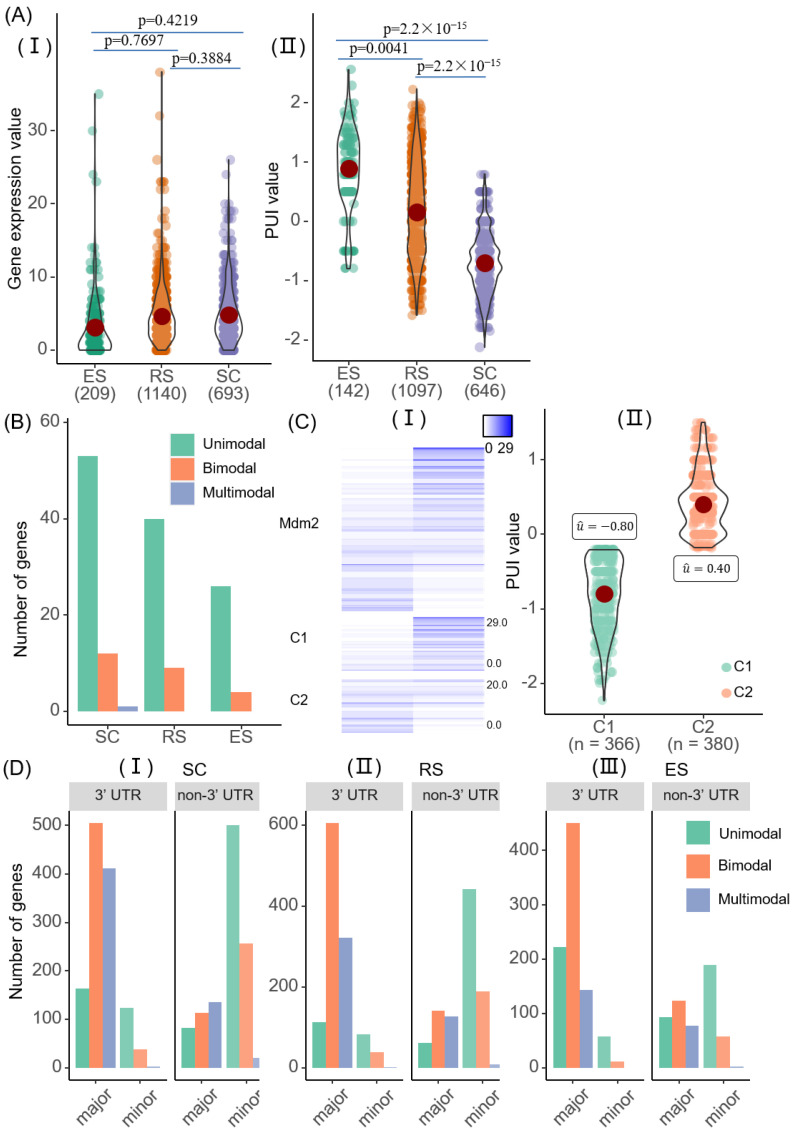
The 3′ UTR APA modality. (**A**) Gene expression (I) and APA usages (II) during *mouse* spermatogenesis differentiation of gene *Mdm2*. (**B**) Modalities identified using scAPAmod for of 3′ UTR-APA genes in each stage. (**C**) Two components of the GMM model (C1 and C2) of the bimodality of gene *Mdm2*. The heatmap of poly(A) site expression (I) and boxplot representation of PUI data (II) of the *Mdm2* gene. The upper part of the heatmap shows the expression of the two poly(A) sites of the gene in different cells. The lower part shows the expression of poly(A) sites belonging to each component of the gene in different cells. Here, the C1 component had 366 cells and the C2 had 380 cells. (**D**) Comparison of modalities identified for major poly(A) sites and minor poly(A) sites in 3′ UTR and non-3′ UTR across stages in SC (I), RS (II) and ES (II) stage.

**Figure 4 ijms-23-08123-f004:**
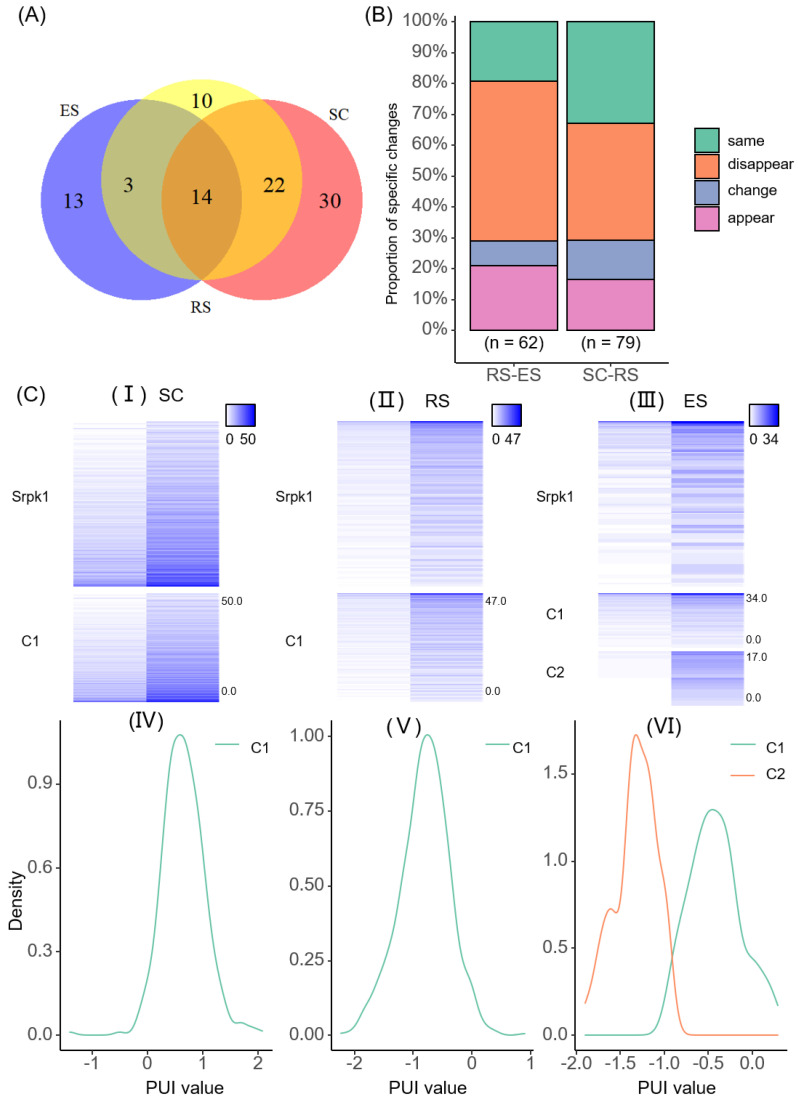
Cell-type-specific modality. (**A**) Statistics on the number of usage patterns genes across different cell types; (**B**) statistics on usage patterns changes of different cell differentiation stages; (**C**) usage pattern changes in *Srpk1* gene across different cell types. (I–III) is expression distribution heat maps of the two poly(A) sites of *Srpk1*, and the (IV–VI) is the density distribution curve of PUI data of *Srpk1*.

**Figure 5 ijms-23-08123-f005:**
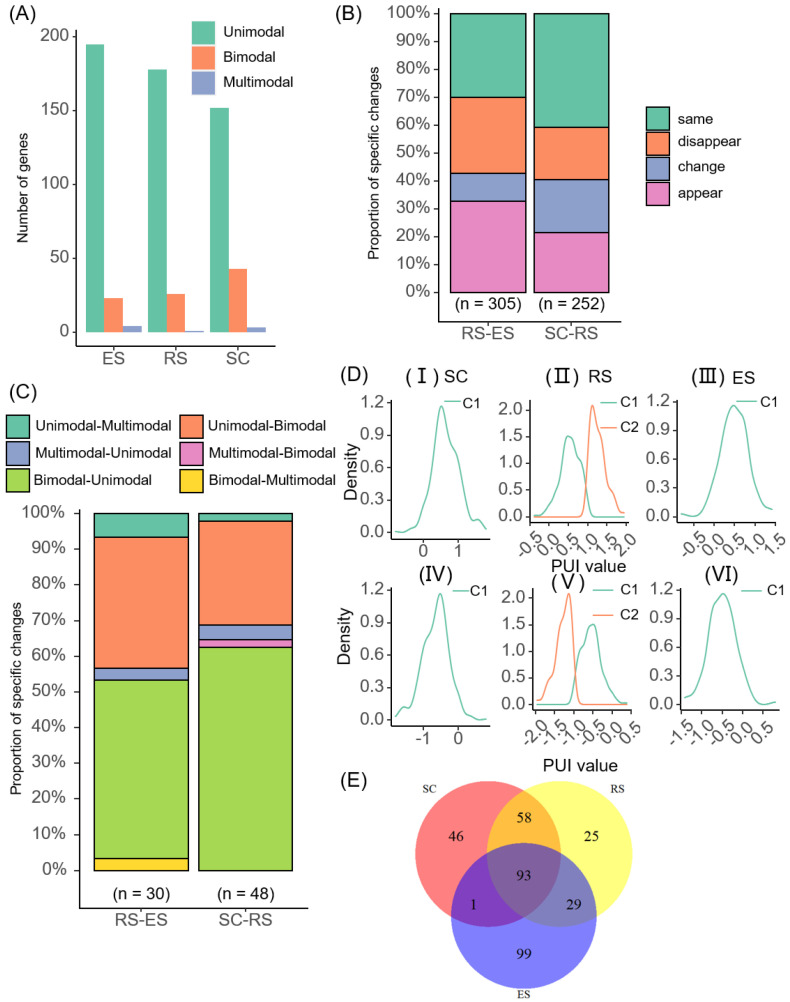
Non-3′ UTR APA modality. (**A**) Statistics on the number of poly(A) sites with detectable usage patterns in different cell types of non-3′ UTR; (**B**) statistics on changes in poly(A) sites with detectable usage patterns of non-3′ UTR cells in the two differentiation stages; (**C**) statistics on changes in poly(A) sites usage patterns of non-3′ UTR in the two differentiation stages; (**D**) the density distribution of proximal (I–III) and the distal (IV–VI) poly(A) site of *Crem* in different cell types; (**E**) statistics on the number of poly(A) sites with detectable usage patterns corresponding to genes with non-3′ UTR APA in different cell types.

## Data Availability

https://github.com/BMILAB/scAPAmod (accessed on 5 June 2022).

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
