# Peer review of "scAPAmod: Profiling Alternative Polyadenylation Modalities in Single Cells from Single-Cell RNA-Seq Data"

_ijms, 2022, doi:10.3390/ijms23158123_

Round 1

Reviewer 1 Report

Lingwu Qian et al. present a new tool, named scAPAmod, that profiles alternative polyadenylation modalities in single cell sequencing data. The main features is the identification of dynamic changes of the pattern of APA usage at the single-cell level. The method has been tested on simulated data and a publicly available single-cell mouse spermatogenesis data set.

Major comment:

* The results are not understandable without having the information provided in the methods section. I suggest to restructure the manuscript.

* The methods text in 4.2.1 describing the metrices used in the analyses is not well written. The text in the beginning is missplaced. The mean definition in the denominator of equation (1) is not well expressed (is it the mean of all cells or the mean in one gene?). Also which log is used in the denominator. And what is the deinition of 'proximal poly(A) site'?

* What is the relationship between PUI_i and ratio_i. What is the motivation to use PUI_i at some instances in the analysis and ratio_i in others.

* The presented analysis is highly dependent on input data and data preprocessing. In the current manuscript only the references to the respective publications (15,40 and 17) are mentioned in section 4.1. I highly recommend to elaborate on input data and data preprocessing for accurate interpretation of the analysis results. It is not clear from the manuscript how the raw expression data was processed (normalization, transformation) before the usage for scAPAmap. Without knowing about these details, a Figure like S5 suggest that no careful normalization and transformation  has been done as genes of large expression have low variance and genes of low expression have high variance.

* In section 4.3 why do you log2 transform the ratio values by adding pseudocount 1 when by definition major and minor sites have count one? Or is it allowed that minor site has count 0? Only major and minor APA site are defined but what about genes with more than two APA sites?

* Fig. 2: In Fig. 2A please show the expression in log scale. Also provide a significance test with null hypothesis of no difference. What are the numbers in brackets in the legend of the x-axis. In Fig. 2C the middle and right plot are redundant (I suggest to remove the density plot). For Mdm2 how many cells are in C1 and C2 respectively and how many cells are their in total? Your interpretation of Fig. 2B is: "During the cell differentiation process from SC to ES, the number of genes with modality of APA usage are gradually decreased (Figure 2B)." Is this not just the consequence of your filtering steps of cells and polyA sites used for the analysis as described in section 4.3. I think it is known that cells of earlier developmental stages have more expressed genes.

* Fig. 3: I am concerned that the number of genes included in the study is very low (numbers in the Venn diagram in panel A).

* Why are you analysing APA sites in  3' UTR and non-3' UTR region separately? Why not analyzing them at the same time?

Minor comments:

* some sentences in abstract and intro are just repeated, see example below. This example is also hard to read as it is a very long sentence.
abstract: However, existing studies that in-
vestigated APA in single cells were either confined to a few cells, or focused on profiling APA dy-
namics between cell types or identifying APA sites, the diversity and pattern of APA usages on a
genomic scale in single cells, especially in seemingly homogeneous cell populations, remains unap-
preciated.
  intro: These pioneering studies focused on profiling differential APA
usages between cell types or identifying APA sites, the diversity and pattern of cell-to-cell
APA usages on a genomic scale among individual cells, especially in seemingly homoge-
neous cell populations, remains unappreciated.

* I would suggest to remove results/conclusions from the Introduction section.

* Abbreviations should be defined at firest occurrence, e.g., PUI.

* Fig. S4: I'm not sure about the usage of feature 'exon'. Does not the features '5UTR' and 'CDS' already comprise all exons of a gene excluding 3'-UTRs?

* Fig. 4D: There are two rows but only one gene, Crem, is described in caption text.

Author Response

Point 1: The results are not understandable without having the information provided in the methods section. I suggest to restructure the manuscript.

Response 1: We apologize for the previous lack of clarity. We have added a new section “Overview of scAPAmod “in Results and updated the schema figure to describe more clearly the pipeline of our method (Figure 1).

Point 2: The methods text in 4.2.1 describing the metrices used in the analyses is not well written. The text in the beginning is missplaced. The mean definition in the denominator of equation (1) is not well expressed (is it the mean of all cells or the mean in one gene?). Also which log is used in the denominator. And what is the deinition of 'proximal poly(A) site'?

Response 2: Thanks for the comment. It is the mean of poly(A) sites in a gene not a cell, and we have revised the text. We have also added the defination of proximal site in “4.2. Calculation of APA index”: “For APA genes containing more than two 3’ UTR poly(A) sites, the one in the proximity of the stop codon is the proximal site and the one closest to the 3’ UTR end is the distal one.”. Log2 is used and we have updated the main text.

Point 3: What is the relationship between PUI_i and ratio_i. What is the motivation to use PUI_i at some instances in the analysis and ratio_i in others.

Response 3: We apologize for the previous lack of clarity. In this study, we identify APA modalities for different kinds of APA genes or APA sites, including 3’ UTR APA genes, non 3’UTR sites and major/minor sites. PUI is an index to quantify the APA usage of an APA gene in single cells, which has been adopted in the previous study for single-cell APA analysis [1]. PUI can be calculated for each APA site to quantify its relative usage. For an APA gene, the PUI of the proximal site was used to represent the APA usage (i.e., 3’UTR length) of the gene, which is to compare the expression of the proximal site with the geometric average of both the proximal and distal sites. However, for major and minor APA sites, we used ratio instead to quantify the usage of the major or minor site. This is because that the major and minor site can be any poly(A) site in an APA gene, if we calculate the PUI value rather than the ratio, then the PUI of the major site will always be high and the minor low. Consequently, for genes have two poly(A) sites, and the APA modality of these genes based on PUI values would always be the same (unimodal). Therefore, we used ratio instead to calculate the usage of the major/minor site.

Accordingly, we have revised the “4.4. Identification of 3’ UTR-APA modality” section to describe more clearly the choice of different indexes.

Point 4: The presented analysis is highly dependent on input data and data preprocessing. In the current manuscript only the references to the respective publications (15,40 and 17) are mentioned in section 4.1. I highly recommend to elaborate on input data and data preprocessing for accurate interpretation of the analysis results. It is not clear from the manuscript how the raw expression data was processed (normalization, transformation) before the usage for scAPAmap. Without knowing about these details, a Figure like S5 suggest that no careful normalization and transformation has been done as genes of large expression have low variance and genes of low expression have high variance.

Response 4: The scRNA-seq dataset used in this study is the same as the one used in previous studies [1-3]. Since the study of scAPAtrap [3] have analyzed this dataset, we followed the same pipeline provided in its GitHub website for data preprocessing and poly(A) site identifcation. No normalization and transformation was performed. We have revised the sentence to avoid confusion: “We used scAPAtrap [3] to identify and quantify poly(A) sites in each single cell, following the same pipeline as in the study of scAPAtrap for data preprocessing and poly(A) site identification…. The expression level represented by the count of unique molecular identifiers (UMIs) of each poly(A) site was also obtained.”. As to the Figure S5, because we used ratio or PUI instead of expression levels (or UMI counts) to quantify the APA usage, we anticipated that the result is irrelvent of the absolute expression value of the gene.

Point 5: In section 4.3 why do you log2 transform the ratio values by adding pseudocount 1 when by definition major and minor sites have count one? Or is it allowed that minor site has count 0? Only major and minor APA site are defined but what about genes with more than two APA sites?

Response 5: Thanks for the comment. The calculation of major/minor is only for genes with at least two poly(A) sites, and only one major and minor site will be obtained for a gene. If the gene has more than two sites, the one with the highest and the lowest expression is selected as the major and minor, respectively. Yes, it is possible that the ratio of a minor site can be 0 in some cells if the site is not expressed in that cell. The reason for adding 1 when performing log2 conversion on the ratio value is to keep the value range of ratio and the value range of PUI in the same order of magnitude. Accordingly, we have revised “4.4. Identification of 3’ UTR-APA modality” to describe the calculation of ratio more clearly.

Point 6: Fig. 2: In Fig. 2A please show the expression in log scale. Also provide a significance test with null hypothesis of no difference. What are the numbers in brackets in the legend of the x-axis. In Fig. 2C the middle and right plot are redundant (I suggest to remove the density plot). For Mdm2 how many cells are in C1 and C2 respectively and how many cells are their in total? Your interpretation of Fig. 2B is: "During the cell differentiation process from SC to ES, the number of genes with modality of APA usage are gradually decreased (Figure 2B)." Is this not just the consequence of your filtering steps of cells and polyA sites used for the analysis as described in section 4.3. I think it is known that cells of earlier developmental stages have more expressed genes.

Response 6: Thanks for the suggestion. We have performed wilcox.test between any two groups of data in Fig. 2A (Figure 3A in the updated manuscript) and added p-values on the plot. The numbers in brackets in the legend of the x-axis indicate the numbers of cells in each group. In Fig 2C (Figure 3C in the updated manuscript), the C1 component of gene Mdm2 has 366 cells and the C2 has 380 cells. Accordingly, we have revised thoroughly Figure 2 (Figure 3 in the updated manuscript) and its legend. Indeed, as the reviewer pointed out, the number of genes is related with the developmental stage, so we have deleted this sentence to avoid making inappropriate conclusions.

Point 7: Fig. 3: I am concerned that the number of genes included in the study is very low (numbers in the Venn diagram in panel A).

Response 7: Indeed, the number of genes identified with APA modality is relatively low in this study, compared to many other single-cell studies. However, our study focused on the APA modality rather than gene expression or APA dynamics analyzed in other studies. We anticipated that genes with APA modalities are less than genes with dynamic APA usages between cell types. This may be due to the following reasons. First, in many other single-cell studies on APA, cells from the same cell type were pooled for the analysis of APA site switching between cell types. But APA modality is identified from heterougeous single cells, and the expression (or APA) profile of an individual cell is much sparser than the pooled cell pupulations. Second, we used the GMM model to identify APA modalities, which require that the APA usages in subpopulations present distinct components. Therefore, our results are conserved and the number of genes with APA modality may be underestimated. It is true that it may not be significant to make conclusions about the trend across development from these genes, however, users can inspect each single gene with APA modality identified by our method and may find genes with biological importance. We have revised the Discussion to discuss this potential limitation of our method.

Point 8: Why are you analysing APA sites in 3' UTR and non-3' UTR region separately? Why not analyzing them at the same time?

Response 8: Yes, genes with 3’ UTR APA or non-3’UTR APA can be analyzed at the same time using our method. Majority of studies on APA focused only on 3’UTR APA sites, while only a small number of studies focused on non-3’UTR sites espetially intronic sites. As these two groups of APA sites represent two different aspects of APA regulation, we chose to describe the results of 3’ UTR APA and non-3’UTR APA separately.

Point 9: some sentences in abstract and intro are just repeated, see example below. This example is also hard to read as it is a very long sentence.

abstract: However, existing studies that investigated APA in single cells were either confined to a few cells, or focused on profiling APA dynamics between cell types or identifying APA sites, the diversity and pattern of APA usages on agenomic scale in single cells, especially in seemingly homogeneous cell populations, remains unappreciated.

intro: These pioneering studies focused on profiling differential APA usages between cell types or identifying APA sites, the diversity and pattern of cell-to-cell APA usages on a genomic scale among individual cells, especially in seemingly homogeneous cell populations, remains unappreciated.

Response 9: Thanks for the suggestion. We have revised the relevant sentences.

Point 10: I would suggest to remove results/conclusions from the Introduction section.

Response 10: We have removed results/conclusions from the Introduction section.

Point 11: Abbreviations should be defined at firest occurrence, e.g., PUI.

Response 11: We have revised thoroughly the main text.

Point 12: Fig. S4: I'm not sure about the usage of feature 'exon'. Does not the features '5UTR' and 'CDS' already comprise all exons of a gene excluding 3'-UTRs?

Response 12: The annotation of poly(A) sites is performed according to the genome annotation. Some special types of genes, such as lncRNA, contain exons. Poly(A) sites found in such genes are located in exons. For protein coding genes, only CDS and UTR is used. We have revised the figure legend to avoid confusion.

Point 13: Fig. 4D: There are two rows but only one gene, Crem, is described in caption text.

Response 13: We apologize for the previous lack of clarity. The two lines of Fig. 4D represent the proximal and distal poly(A) site of the gene. We have revised the figure legend.

  1. Shulman, E.D. and R. Elkon, Cell-type-specific analysis of alternative polyadenylation using single-cell transcriptomics data. Nucleic Acids Res, 2019. 47(19): p. 10027-10039. http://www.ncbi.nlm.nih.gov/pubmed/31501864
  2. Lukassen, S., E. Bosch, A.B. Ekici, and A. Winterpacht, Characterization of germ cell differentiation in the male mouse through single-cell RNA sequencing. Sci Rep, 2018. 8(1): p. 6521. https://www.ncbi.nlm.nih.gov/pubmed/29695820
  3. Wu, X., T. Liu, C. Ye, W. Ye, and G. Ji, scAPAtrap: identification and quantification of alternative polyadenylation sites from single-cell RNA-seq data. Brief Bioinform, 2021. 22(4).

Reviewer 2 Report

Lingwu Qian et al submitted the manuscript to IJMS addressing the APA using scRNA-seq. The authors have covered most of the aspects of the research question. There are some minor suggestions i.e.

  1. Some approaches are introduced abruptly without basic information such as BAT-seq  (Line 45) and cTag-PAPERCLIP (Line 49).
  2. The same is for the results part

Author Response

Point 1: Some approaches are introduced abruptly without basic information such as BAT-seq (Line 45) and cTag-PAPERCLIP (Line 49).

Response 1: Thans for the suggestion. We have removed these abbreviations of sequencing protocols to avoid confusion.

Point 2: The same is for the results part

Response 2: We have revised the main text thoroughly to make our manuscript more understandable. Particularly, we have added a new section “Overview of scAPAmod “in Results and updated the schema figure to describe more clearly the pipeline of our method (Figure 1).

Reviewer 3 Report

The authors present a method to investigate APA events in single cell data. According to a recent paper by Yang et al. 2021 BMC Biology, modality of APA patterns could identify subpopulations. The development of a proper method or an analysis framework (as suggested by the authors), would in turn help to decipher the differences between subpopulations, and identify subpopulations based on APA events.

Major Comments:

The authors have made a significant attempt by suggesting a framework: scAPAmod, to understand the patterns for APA usage, but requires more work to prove the usability and validations of their framework.

(i) The authors should describe their whole framework in more detail.  The authors could provide a workflow to aid in describing the important steps and validation experiments etc. for the scAPAmod framework.

(ii) The authors should verify their claims of their suggested framework, and implement their framework towards identifying APA events and distinguishing the subpopulations (from popular and well established, benchmark datasets like human PBMC etc.). How does the identified APA events using scAPAmod, delineate known cell type clusters (may be silhouette for assessment of the clusters etc.)? The authors should also consider comparing with bulk RNA seq datasets to validate their claims.

(iii) Discuss about how well does scAPAmod perform in recovering missing APA events caused by low amounts of APA events from benchmark scRNA seq datasets as well as missing data in simulations ?

(iv) Regarding the simulation data, I would suggest using standard methods to generate and providing more details. Moreover, generating simulated data over 100 cells is not enough. Also, the authors should show that their framework with GMM scales well with higher number of cells, such as 10000 (with newer technologies). The authors should also explain how the missing data was imputed to infer modalities.

(v) There are many terms that are not explained properly. Some are:

  • The authors should provide more explanation regarding ‘modality’. The paper by Yang et al. 2021 that the authors have referred while discussing about modalities, uses 5 modalities (a) distal (b) proximal (c) bimodal (d) middle (e)  multimodal, while the authors use 3 modalities. How are these different ? How does the discovered modalities compare within the subpopulations ? Is it more improved with the other methods such as scDaPars / SAPAS etc. An example gene with the poly A sites on the isoforms and read coverage along with different modalities would help to understand the concepts.
  • The definition of ‘components’ (mentioned first in line: 149) is missing.

(vi) Authors use ISOP, published in (Bioinformatics 2018), but do not explain thoroughly the reason behind the choice of the method.  ISOP is introduced first in the text in line 118 , but authors mention about it in line 511. Authors say that “ISOP …..  can in principle be used to identify different patterns of APA in a group of cells“ as the reason for using in the comparison. The authors should include comparisons to other methods that are capable of detecting APA events such as scDaPars (Gao et al. Genome Research 2021) and SAPAS (Yang et al. BMC Biology 2021), scAPA (Shulman and Elkon NAR 2019), Sierra (Patrick et al. Genome Biology 2020) etc.

Minor Comments:

(i) The authors use scAPAtrap to identify and quantify poly(A) sites in each single cell (both in the 3UTR as well as non 3UTR region). But the reason for the specific choice of this tool is not discussed. Moreover, the preprocessing steps for the scRNA seq data from the samples is also missing.

(ii) Sometimes it is difficult to understand what the author is trying to say. Such as:

  • The sentence structure is not proper : line 416

  • too long sentence: difficult to understand line 60 – 63

  • The equations are not explained properly: line 431

  • The last few lines of the abstract (line 21 ...) onwards is exactly similar to the last paragraph of the introduction (line 100 ...)

Author Response

Point 1: The authors should describe their whole framework in more detail. The authors could provide a workflow to aid in describing the important steps and validation experiments etc. for the scAPAmod framework.

Response 1: We apologize for the previous lack of clarity. We have added a new section “Overview of scAPAmod “in Results and updated the schema figure to describe more clearly the pipeline of our method (Figure 1).

Point 2: The authors should verify their claims of their suggested framework, and implement their framework towards identifying APA events and distinguishing the subpopulations (from popular and well established, benchmark datasets like human PBMC etc.). How does the identified APA events using scAPAmod, delineate known cell type clusters (may be silhouette for assessment of the clusters etc.)? The authors should also consider comparing with bulk RNA seq datasets to validate their claims.

Response 2: Thanks for the suggestion. As suggested by the reviewer, we also used scAPAmod to analyze a Human PBMC 4k data (https://support.10xgenomics.com). The gene-cell expression matrix of this dataset contains 33694 genes in 4340 cells. After identification of poly(A) sites from the raw scRNA-seq using scAPAtrap, 15304 poly(A) sites were identified in 3’ UTRs, and 3170 3’ UTR-APA genes were obtained. We considered three main cell types for identifying APA modalities: T cells (2308 cells), B cells (612 cells), and Monocytes (1164 cells). Only a small number of APA genes with modalities were detected -- 13 genes in T cells, 11 genes in B cells and 17 genes in Monocytes cells. Among these genes, unimodal and bimodal were detected, while no multimodal was detected. The relatively low number APA modalities present in the PBMC data may be due to the following reasons. First, the APA profile obtained from the PBMC data is sparse, even sparser than the mouse spermatogenesis data we analyzed, consequently, there are not sufficient cells for modality identification. Second, the APA dynamics among cell types in PBMCs is not as distinguishable as during the spermatogenesis, or the impact of APA regulation in PBMCs is not as significant as that during the spermatogenesis. Third, we used the GMM model to identify APA modalities, which require that the APA usages in subpopulations present distinct components. Therefore, our results are conserved and the number of genes with APA modality may be underestimated. Due to the limited number of genes with APA modalities obtained from the PBMC data, we did not perform downstream analysis for the PBMC data. However, we added a paragraph in the Discussion to discuss about the potential limitation of our method.

As suggested by the reviewer, to examine whether genes with APA modalities can delineate known cell type clusters, we used Seurat V4.0.2 (dim.use=4) to cluster cell types based on the APA expression profile of genes with APA modalities. A total of 420 genes with 3’UTR-APA or non 3’UTR-APA modalities were used. The mean silhouette width and ARI score of the clustering result based on the APA profile is 0.71 and 0.92, respectively (Figure S11A). In contrast, the mean silhouette width and ARI score of the clustering result based on gene expression profile of all genes is 0.60 and 0.67, respectively (Figure S11B). Result based on genes with APA patterns is not only higher than that based on whole gene expression profiles, but the feature space for clustering is greatly reduced (420 vs. 2042 genes). This result preliminarily suggests that the APA expression profile of genes with APA modalities is sufficient to distinguish cell types and indicates the potential use of our scAPAmod tool for identifying important features for cell type clustering. We have added this preliminary result in Discussion to demonstrate the potential use of our tool for improving cell type clustering.

We fully agree with the reviewer that it should be more convincing if we could use the RNA-seq datasets for validation. However, we are afraid that it is not able to use the bulk RNA-sesq data for verification of APA modalities at present. This is because that APA modality represents the pattern of the cell-cell heterogeneity of APA usages among a cell population, which can't be inferred from bulk RNA-seq data. Indeed, there are many studies compared APA dynamics or differential APA usages between bulk and single-cell RNA-seq, however such comparison is not feasible for APA modalities. Because APA modality reflects cell-cell heterogeneity, which is different from APA dynamics that considers cell types but not individual cells.

Point 3: Discuss about how well does scAPAmod perform in recovering missing APA events caused by low amounts of APA events from benchmark scRNA seq datasets as well as missing data in simulations ?

Response 3: Thanks for the suggestion. We did not impute missing data but remove cells with missing values before identifying modalities. This is because that there are no dropout imputation methods available for imputation APA signals. Although a recent tool, scDaPars [1], was proposed for identifying and imputing APA signals from scRNA-seq, the scDaPars pipeline identified APA signals using DaPars which is not very scalable to scRNA-seq data. Therefore, we did not consider imputation in this study. However, scAPAmod is flexible to incorporate the imputation step into its framework when dedicated imputation tools are available for recovering APA signals from scRNA-seq. We have revised the Discussion to add this point.

Point 4: Regarding the simulation data, I would suggest using standard methods to generate and providing more details. Moreover, generating simulated data over 100 cells is not enough.  Also, the authors should show that their framework with GMM scales well with higher number of cells, such as 10000 (with newer technologies). The authors should also explain how the missing data was imputed to infer modalities.

Response 4: Thanks for the suggestion. As in the response to the above comment, we did not impute missing data but remove cells with missing values before identifying modalities. To further evaluate the scalability of scAPAmod for higher number of cells, we constructed simulation datasets with 8000 cells using the same data generation procedure as the 100-cell simulation data. Compared to the results on the 100-cell simulation data, we found that the recognition performance of both scAPAmod and ISOP decreased in the 8000-cell noise-free simulation data (Figure S2A vs. Figure 2A). The performance of scAPAmod is still significantly better than ISOP (Figure S2). With the increase of the proportion of missing values, the number of falsely identified modalities by either scAPAmod or ISOP increased (Figure S2B). When the simulated noise-free data is added with 5% to 60% missing values, less than 100 multimodal items were falsely predicted as bimodal by scAPAmod. In contrast, ISOP predicted wrongly more unimodal and multimodal items as bimodal. This result demonstrates that scAPAmod can mitigate the impact of high dropout rate and can effectively identify true modalities even for sparse data with high sample size.

We calculated the ARI value on different degrees of noise (0%, 5%, 10%, 15%, 20%, 25% and 30%) (Figure S2C). For bimodal items, ARI scores of scAPAmod and ISOP are both very close 1, which means that both methods can accurately identify the bimodal from the data. In contrast, for multimodal items, ARI scores of both scAPAmod and ISOP are not high. For bimodal items, ARI scores of scAPAmod and ISOP were very close to 1, which is essentially the same as the results of the 100-cell simulation data. For the multimodal items, ARI scores of scAPAmod and ISOP were much smaller than the results from 100-cell simulation data. Finally, we simulated the bimodal data containing five different ratios of cells number of two components. Result shows that both scAPAmod and ISOP can identify correctly the bimodal pattern with varied cell ratio of the two components (Figure S3E and Figure S1E) , which is essentially the same as the results of the 100-cell simulation data. These results demonstrate the high scalability and performance of scAPAmod on data with different sample sizes and degrees of noises.

Accordingly, we have added these results in “2.2.Evaluation of scAPAmod for APA modality identification” of the main text.

Point 5: There are many terms that are not explained properly. Some are:

The authors should provide more explanation regarding ‘modality’. The paper by Yang et al. 2021 that the authors have referred while discussing about modalities, uses 5 modalities (a) distal (b) proximal (c) bimodal (d) middle (e)  multimodal, while the authors use 3 modalities. How are these different ? How does the discovered modalities compare within the subpopulations ? Is it more improved with the other methods such as scDaPars / SAPAS etc. An example gene with the poly A sites on the isoforms and read coverage along with different modalities would help to understand the concepts.The definition of ‘components’ (mentioned first in line: 149) is missing.

Response 5: We apologize for the previous lack of clarity. We have added a section “Overview of scAPAmod” in Results to describe more clearly our pipeline and explain many terms, such as modality and component. We also updated Figure 1 to add intuitive examples for different modalities.

Yes, Yang et al. 2021 analyzed APA dyanamics in GABAergic neuron types and also identified modalities of APA sites. They first defined reference binned distributions of APA usages, then compared each gene’s binned distribution of distal poly(A) site usage to the reference to determine the gene’s modality. We anticipate that the proximal, distal and middle modalities of Yang et al. 2021 are corresponding to the unimodel in scAPAmod. The bimdal and multimodal are corresponding to the bimodal and multimodal of scAPAmod, respectively. They explicitly define the desired distribution of different modalities, e.g., distal = c(0,0,1), proximal = c(1,0,0), middle = c(0.1,0.8,0.1), etc, and selected the modality of the closest reference distribution as the modality of the gene’s poly(A) site usage. Consequently, they could not recognize modalities beyond defined reference, and may assign modality for genes that actually do not present a significant modality. In Yang et al., they only analyzed the modalities of 3’ UTR-APA. In contrast, scAPAmod is more flexible without determining any reference distribution, and can be applied to the identification of modalities of both 3’UTR and non 3’UTR-APA. Moreover, with the statistically significant components inferred by the GMM model used in scAPAmod, we could obtain the mean value of each component (e.g., mean PUI) and the corresponding cells for component, which could then determine the preferential usage of each APA site in different cell subpopulations. As suggested by the reviewer, we have included SAPAS for comparison, and results show that scAPAmod outperforms both ISOP and SAPAS. Detailed results are presented in “Evaluation of scAPAmod for APA modality identification”.

Although methods such as scDaPars also investigated APA in single cells, they focused on profiling APA dynamics between cell types or identifying APA sites, which is different from the identification of APA modalities in this study. APA modality represents the pattern of the cell-cell heterogeneity of APA usages among a cell population, which is different from APA dynamics that considers differential use of APA sites between two cell populations (e.g., cell types) or two cells. Therefore, it is not appropriate to compare scAPAmod with scDaPars.

Accordingly, we have revised the Results to add the new results and the Discussion to discuss further these issues.

Point 6: Authors use ISOP, published in (Bioinformatics 2018), but do not explain thoroughly the reason behind the choice of the method.  ISOP is introduced first in the text in line 118, but authors mention about it in line 511. Authors say that “ISOP …..  can in principle be used to identify different patterns of APA in a group of cells“ as the reason for using in the comparison. The authors should include comparisons to other methods that are capable of detecting APA events such as scDaPars (Gao et al. Genome Research 2021) and SAPAS (Yang et al. BMC Biology 2021), scAPA (Shulman and Elkon NAR 2019), Sierra (Patrick et al. Genome Biology 2020) etc.

Response 6: Thanks for the suggestion. This study focused on the APA modality rather than gene expression or APA dynamics analyzed in many other single-cell studies [11, 15-17]. Although methods such as scDaPars, scAPA and Sierra also investigated APA in single cells, they focused on profiling APA dynamics between cell types or identifying APA sites, which is different from the identification of APA modalities in this study. APA modality represents the pattern of the cell-cell heterogeneity of APA usages among a cell population, which is different from APA dynamics that considers differential use of APA sites between two cell populations (e.g., cell types) or two cells.

We included ISOP for comparison is that it is the only method for analyzing cell-to-cell variability at the isoform-level when we developed our scAPAmod. We anticipated that an APA site can be regarded as an isoform, which thus can be analyzed by ISOP. As suggested by the reviewer, we have included SAPAS for comparison, and results show that scAPAmod outperforms both ISOP and SAPAS. Detailed results are presented in “Evaluation of scAPAmod for APA modality identification”.

Point 7: The authors use scAPAtrap to identify and quantify poly(A) sites in each single cell (both in the 3UTR as well as non 3UTR region). But the reason for the specific choice of this tool is not discussed. Moreover, the preprocessing steps for the scRNA seq data from the samples is also missing.

Response 7: Thanks for the comment. We have updated the new flow chart which can be more intuitive to see our preprocessing steps. Moreover, we calculated the expression of poly(A) sites rather than the traditional genes expression genes, so the normalization method for genes is not suitable for normalizing the expression of poly(A) sites. But since we calculated the PUI or ratio, which is independent of the size of the gene expression level, there is no need for normalization.

Point 8: Sometimes it is difficult to understand what the author is trying to say. Such as:

The sentence structure is not proper : line 416

too long sentence: difficult to understand line 60 – 63

The equations are not explained properly: line 431

The last few lines of the abstract (line 21 ...) onwards is exactly similar to the last paragraph of the introduction (line 100 ...)

Response 8: We have fixed these minor issues above and removed the last paragraph of the introduction.

  1. Gao, Y., L. Li, C.I. Amos, and W. Li, Analysis of alternative polyadenylation from single-cell RNA-seq using scDaPars reveals cell subpopulations invisible to gene expression. Genome Research, 2021. 31(10): p. 1856-1866. http://genome.cshlp.org/content/early/2021/05/25/gr.271346.120.abstract

Round 2

Reviewer 3 Report

The authors have dedicatedly improved on the paper based on the previous review comments. But its still not enough. A through revision of the work is required in the following aspects:

(i) The authors should focus more on the novelty of the work (in the introduction and discussion) as well as the limitations of their work. Some terms that have been used repeatedly like patterns, components, modalities , whose proper definition is missing.

(ii) As this is a method centric paper, the authors should try and comment on other scRNAseq datasets as well (for example, the scRNAseq datasets used in Gao et al. Genome Research 2021). The authors should thereby try to validate some of the results (or provide additional insights) discussed in previous attempts to uncover APA events.

(iii) Moreover, the authors should also make serious effort to improving the english, and hence to improve the readability of the manuscript. As of now, its difficult to comprehend. The terms used and their definitions are not in proper order. Some paragraphs of the section are really long. The paragraphs should begin with a general statement that defines the general theme in the entire paragraph. In case the paragraph has mixed results, providing paragraph title is recommended. 

(iv) The authors should also put proper effort to refer and cite figures. They should also make sure the figures have proper axes labels, subplot labels, and proper explanation of the results in the text as well as in the figure legend.

The following concerns in the manuscript should be addressed.

Major Comments:

1. The authors have completely removed the last paragraph of the introduction (from version 1 of the draft). I think the last paragraph should highlight in brief the shortfalls of the previous attempts (if any), novelty of the work/approach, the approach of the scAPAmod framework (in brief) and the key.

2. The workflow provided in Figure 1 is really helpful to understand the processing steps in determining the APA modalities. But, (a) the steps as shown in the figure, have not been explained in details in the materials and methods.

(b) A part of the figure 1 seems to be taken directly from Sierra (Patrick et al.2020), without proper acknowledgement (also identifiable with the lower resolution than the rest of the figure) to the figure. The authors should redraw this.

(c) Moreover, what is the difference between Gene1.1 labelled in Poly(A) site matrix and Gene 1 labelled in APA signal matrix?

Also, in the legend of fig 1, it would be good to mention the step number and explain in brief the steps of the workflow in very brief.

2. In Fig 2B: the difference between the subplots is not quite visible (in the number of different modalities detected) with varying missing values. Moreover, a bar plot is not the best way to compare the number of modalities detected across a range of missing values. I think one should also show the standard deviation (with error bars) the number of different modalities detected. Moreover, please mention in more details the specifics of the simulation used in the study.

3. line 132 says “Most patterns identified by ISOP are II pattern (37.6%, or bimodal) or V pattern (32.33%), with 133 much less patterns being I pattern (13.4%, or unimodal), VI pattern (9.8%) or XI pattern 134 (6.87%)”. The authors should define the ‘patterns’ or refer/cite their definition. Moreover, the definition of ‘components’ w.r.t modalities is vague. A proper definition (may be in the Materials and Methods section) is necessary.

In the fig. 3 legend (Line 248) says: “Here the C1 component has 366 cells and the C2 has 380 cells ”, what does it mean for a component to have 366 cells ? Its confusing.

4. No reference in text for figures S3A, S3B, S3C, S3D. I did not understand figure S3C .

5. It would be nice to have some biological insights or association with biological processes for the genes that are mentioned (that show different modalities or shortening or lengthening), that could be linked to spermatogenesis. IGV browser snapshot (as in Fig 5 , Yang et al. BMC Biology 2021) of variation in the read coverage in 3UTR / non-3UTR events in some of the genes mentioned can be helpful to highlight the findings of the scAPAmod findings.

6. As also commented in the first version, the authors should definitely try their approach (re: PBMC dataset) on other scRNAseq datasets (such as mentioned in Gao et al. Genome Research 2021), and discuss in details if their method implementation succeeds (thus providing more insights and inferences to results) or fails (reasons of the failure). Also, comments on the limitations of scAPAmod in the discussion is missing.

7. The paper does not state any thresholds (in PUI or proportion or any significance test etc.) used to determine the tabular results or the selection of the specific genes. Please discuss the thresholds used to identify the genes (in paragraph from line 227) such as Mdm2, Rpl34 etc. 

8. Based on the PUI_i (defined in eqn (1)), can the authors also infer the shortening or lengthening of the 3UTR? the term ‘pui’ used in equation (3) is not defined.

Minor Comments

1. “the scDaPars pipeline identified APA dynamics using DaPars which is not very scalable to scRNA-seq data.” The authors should mention any proof, reason or reference for such a claim

2. The silhouette plot as referred in the manuscript  (fig S11A, B), is missing 

3. section 4.1: for ease of reading , the authors should include the technology of the scRNAseq data, and source. Also, describe in brief the preprocessing steps

4. paragraph from line 553: The paragraph seems to be defining the steps for an EM algorithm, and other thresholds used. However, the steps in the text is complicated to understand. May be a pseudocode, (incl. the constraints) with the EM algorithm, the expectation step and the maximisation step would help to understand the computational complexity.

5. line 579 : what is “expression level of APA site”?

6. line 215: “Almost all cells (2028/2042, 99%) are …...”, what does this mean ? 

Other small comments:

1. twice use of section number 2.2

2. line 158 : missing reference for the ARI

line 208: missing reference to the dataset.

line 211: missing bracket

line 229, 231: instead of referring the subfigure as ‘left’, give roman numerals

3. line 488, 489: what did the authors mean by “special or specific functions”. More specific detaiks required.

fig 2C: missing x axis label

fig. 3D: missing legend label

line 567: missing reference to clusterR package.

4. As of now, the paper is difficult to comprehend. Some examples of improper english usage:

ln 489, 490: sentences beginning with ‘and’.

ln 469: sentence not structured properly. Meaning not clear

Author Response

Point 1: The authors have completely removed the last paragraph of the introduction (from version 1 of the draft). I think the last paragraph should highlight in brief the shortfalls of the previous attempts (if any), novelty of the work/approach, the approach of the scAPAmod framework (in brief) and the key.

Response 1: Thanks for your advice. We have supplemented the last paragraph of the introduction.

Point 2: The workflow provided in Figure 1 is really helpful to understand the processing steps in determining the APA modalities. But, (a) the steps as shown in the figure, have not been explained in details in the materials and methods.

(b) A part of the figure 1 seems to be taken directly from Sierra (Patrick et al.2020), without proper acknowledgement (also identifiable with the lower resolution than the rest of the figure) to the figure. The authors should redraw this.

(c) Moreover, what is the difference between Gene1.1 labelled in Poly(A) site matrix and Gene 1 labelled in APA signal matrix?

Also, in the legend of fig 1, it would be good to mention the step number and explain in brief the steps of the workflow in very brief.

Response 2: Thanks for your suggestions. (a) We have modified the figure to add the step number. We have described steps of the figure in the figure legend and cited the figure in the main text.

(b) We have redrawn this part of Figure 1.

(c) In the previous figure, there was no relationship between the gene1 in the two matrices. In the updated figure, we have linked the gene 1 in the two matrices, now the gene 1 in the APA matrix is the ratio of the proximal site in the poly(A) site matrix.

Point 3: In Fig 2B: the difference between the subplots is not quite visible (in the number of different modalities detected) with varying missing values. Moreover, a bar plot is not the best way to compare the number of modalities detected across a range of missing values. I think one should also show the standard deviation (with error bars) the number of different modalities detected. Moreover, please mention in more details the specifics of the simulation used in the study.

Response 3: Thanks for your suggestions. We have modified Figure 2 A and B to show more clearly the number of error recognitions among the three tools. Now, Figure 2A and B shows the number of recognition errors for different methods. The generation of the simulated data was described in Materials and Methods. We have added “see Materials and Methods” in the result.

Point 4: Line 132 says “Most patterns identified by ISOP are II pattern (37.6%, or bimodal) or V pattern (32.33%), with 133 much less patterns being I pattern (13.4%, or unimodal), VI pattern (9.8%) or XI pattern 134 (6.87%)”. The authors should define the ‘patterns’ or refer/cite their definition. Moreover, the definition of ‘components’ w.r.t modalities is vague. A proper definition (may be in the Materials and Methods section) is necessary.

In the fig. 3 legend (Line 248) says: “Here the C1 component has 366 cells and the C2 has 380 cells ”, what does it mean for a component to have 366 cells ? Its confusing.

Response 4: Thanks for your suggestions. As described in section 2.1, ‘patterns’ and ‘modalities’ are equivalent. We have introduced ‘components’ in section 2.1. In addition, ‘component’ is the concept of Gaussian mixture model, which is defined in section 4.3.

We used scAPAmod to identify the gene Mdm2 as bimodal which means that the PUI data of all cells of this gene obeyed two Gaussian distributions (two components). According to the identification results, the PUI of 366 cells belonged to first component (C1), and the PUI of 380 cells belonged to second component (C2).

Point 5: No reference in text for figures S3A, S3B, S3C, S3D. I did not understand figure S3C.

Response 5: Figure S3C shows that when the two components of the bimodal have different proportions of cell numbers, ISOP recognizes them all as V patterns. We have modified the legend of Figure S3C to explain the figure more clearly.

Point 6: It would be nice to have some biological insights or association with biological processes for the genes that are mentioned (that show different modalities or shortening or lengthening), that could be linked to spermatogenesis. IGV browser snapshot (as in Fig 5 , Yang et al. BMC Biology 2021) of variation in the read coverage in 3UTR / non-3UTR events in some of the genes mentioned can be helpful to highlight the findings of the scAPAmod findings.

Response 6: Thank you for the suggestion. We have tried our best to search the gene function of these genes, and mentioned their biological functions in the main text. Fig 5 of Yang et al. shows the read coverages of cell type specific APA events. However, APA modality characterizes the usages of APA sites in individual cells of a cell population, which cannot be displayed at the cell type level. Fig. 4c in our manuscript intuitively shows usages of APA in single cells for an APA modality. Therefore, we did not show the IGV plot in our manuscript. As a matter of fact, we previously developed a database called scAPAdb, which can view similar plot as Fig 5 of Yang et al. (e.g., for gene Actb which we found an APA modality: http://www.bmibig.cn/scAPAdb/groups/Search/gene_graph.php?geneid=ENSMUSG00000029580&GSM=GSM2803334&Type=pdui&pid=ENSMUSG00000029580).

Point 7: As also commented in the first version, the authors should definitely try their approach (re: PBMC dataset) on other scRNAseq datasets (such as mentioned in Gao et al. Genome Research 2021), and discuss in details if their method implementation succeeds (thus providing more insights and inferences to results) or fails (reasons of the failure). Also, comments on the limitations of scAPAmod in the discussion is missing.

Response 7: Thank you for the suggestion. In our last revision, we have performed additional analysis on the PBMC data. In this version, we have added the results of PBMC in the Discussion to describe in more details the limitation of scAPAmod.

Point 8: The paper does not state any thresholds (in PUI or proportion or any significance test etc.) used to determine the tabular results or the selection of the specific genes. Please discuss the thresholds used to identify the genes (in paragraph from line 227) such as Mdm2, Rpl34 etc.

Response 8: In section 4.4 and section 4.5, we illustrated how we set thresholds to filter cells and genes in the 3’UTR and non-3’UTR regions, respectively. Specifically, we filtered out poly(A) sites expressed in less than a quarter of cells or cells with less than one-tenth of expressed poly(A) sites. Only genes with at least two 3’ UTR poly(A) sites were used for subsequent APA modality analysis. Gene Mdm2 and Rpl34 are the two gene of the result after filtering.

Point 9: Based on the PUI_i (defined in eqn (1)), can the authors also infer the shortening or lengthening of the 3UTR? the term ‘pui’ used in equation (3) is not defined.

Response 9: It cannot infer the shortening or lengthening of 3’UTR based on the PUI of a single poly(A) site. However, we use the PUI of the proximal poly(A) site to characterize the overall usage of a gene, so the length of the 3’UTR can be inferred according to the PUI of a gene. We have already explained it in the third paragraph of section 4.2.

The term ‘pui’ used in equation (3) represents the PUI value of a gene or a poly(A) site, or the ratio of a poly(A) site. For consistency, ‘pui’ is changed to ‘’ in equation (3).

Point 10: “the scDaPars pipeline identified APA dynamics using DaPars which is not very scalable to scRNA-seq data.” The authors should mention any proof, reason or reference for such a claim

Response 10: We tried scDaPars once. However, it is indeed not very scalable to scRNA-seq data. DaPars is a tool for poly(A) site identification from bulk RNA-seq. scDaPars used DaPars for poly(A) site identification from scRNA-seq, which treats each cell as a sample in bulk RNA-seq. As there are hundreds or thousands of cells in a scRNA-seq data, it is not practical to use scDaPars for imputation as it first employs DaPars for poly(A) site identification.

However, to avoid confusion, we have modified the sentence to “the scDaPars pipeline identified APA dynamics using DaPars, the principle of which was different from the tools used in our study, scAPAtrap or Sierra”.

Point 11: The silhouette plot as referred in the manuscript (fig S11A, B), is missing

Response 11: Thanks for your suggestion. We have supplemented these two silhouette plots (fig S11C, D).

Point 12: section 4.1: for ease of reading , the authors should include the technology of the scRNAseq data, and source. Also, describe in brief the preprocessing steps

Response 12: Thanks for your suggestions. The data was from 10x Chromium. We used scAPAtrap to identify and quantify poly(A) sites in each single cell, following the same pipeline as in the study of scAPAtrap for data preprocessing and poly(A) site identification. We have added the technology and cited the scAPAtrap paper in section 4.1.

Point 13: paragraph from line 553: The paragraph seems to be defining the steps for an EM algorithm, and other thresholds used. However, the steps in the text is complicated to understand. May be a pseudocode, (incl. the constraints) with the EM algorithm, the expectation step and the maximisation step would help to understand the computational complexity.

Response 13: The threshold in this paragraph refers to the threshold of the BIC (Bayesian information criterion) value. We determined the number of components based on this threshold. We have provided a pseudocode to aid understanding.

Algorithm: scAPAmod

Input: PUI value of a gene or a poly(A) site, or the ratio of a poly(A) site

Step1. BIC(1), BIC(2), BIC(3)  Optimal_Clusters_GMM(, 3, criterion = "BIC");

Step2. remove the largest BIC value among the three BIC values;

Step3. compare the remaining two BIC values, recorded smaller BIC value as BIC(s) and larger BIC value as BIC(l);

Step4. if

then

else

Step5. for i  1 to

do if the number of cells of i-th component < 10

    then

Output: number of components

Note: BIC(K) represents the BIC value corresponding to the Gaussian mixture model obeying K Gaussian distributions.

Point 14: line 579 : what is “expression level of APA site”?

Response 14: Thanks for your comment. “expression level of APA site” represents the read counts of each APA site of a gene. We have modified the sentence as: “the expression level (i.e., read counts) of APA sites in a gene”.

Point 15: line 215: “Almost all cells (2028/2042, 99%) are …...”, what does this mean ?

Response 15: Thanks for your comment. “Almost all cells (2028/2042, 99%) are …...” means that the vast majority of cells. We have changed to “99% of cells (2028/2042) are …...”.

Point 16: twice use of section number 2.2

Response 16: Thanks for your comment. We have revised the corresponding numbers.

Point 17: line 158 : missing reference for the ARI

line 208: missing reference to the dataset.

line 211: missing bracket

line 229, 231: instead of referring the subfigure as ‘left’, give roman numerals

Response 17: Thanks for the suggestions. We have revised these issues.

We have supplemented the references for ARI and the dataset.

We have supplemented the bracket.

We've used roman numerals instead of words like "left" and "right" to refer to subgraphs.

Point 18: line 488, 489: what did the authors mean by “special or specific functions”. More specific detaiks required.

fig 2C: missing x axis label

fig. 3D: missing legend label

line 567: missing reference to clusterR package.

Response 18: Thanks for your comments. We have revised these issues.

To avoid confusion, we have provided some details removed“special or specific functions”.

We have supplemented the x axis label in fig 2 C and legend label of figure 3 D.

We have supplemented the references for clusterR package.

Point 19: As of now, the paper is difficult to comprehend. Some examples of improper english usage:

ln 489, 490: sentences beginning with ‘and’.

ln 469: sentence not structured properly. Meaning not clear

Response 19: Thanks for your comments. We have revised these issues. We also have tried our best to revise improper expressions in the full text.
